# CS-Isolate: Extracting Hard Confident Examples by Content and Style Isolation

**Yexiong Lin**[1]    **Yu Yao**[2,3]    **Xiaolong Shi**[1]
**Mingming Gong**[4]    **Xu Shen**[5]    **Dong Xu**[6]    **Tongliang Liu**[1*]
[1]The University of Sydney; [2]Mohamed bin Zayed University of Artificial Intelligence;
[3]Carnegie Mellon University; [4]The University of Melbourne;
[5]Alibaba DAMO Academy; [6]The University of Hong Kong.

## Abstract

Label noise widely exists in large-scale image datasets. To mitigate the side effects of label noise, state-of-the-art methods focus on selecting confident examples by leveraging semi-supervised learning. Existing research shows that the ability to extract hard confident examples, which are close to the decision boundary, significantly influences the generalization ability of the learned classifier. In this paper, we find that a key reason for some hard examples being close to the decision boundary is due to the entanglement of style factors with content factors. The hard examples become more discriminative when we focus solely on content factors, such as semantic information, while ignoring style factors. Nonetheless, given only noisy data, content factors are not directly observed and have to be inferred. To tackle the problem of inferring content factors for classification when learning with noisy labels, our objective is to ensure that the content factors of all examples in the same underlying clean class remain unchanged as their style information changes. To achieve this, we utilize different data augmentation techniques to alter the styles while regularizing content factors based on some confident examples. By training existing methods with our inferred content factors, CS-Isolate proves their effectiveness in learning hard examples on benchmark datasets. The implementation is available at https://github.com/tmllab/2023_NeurIPS_CS-isolate.

## 1   Introduction

Large-scale machine learning datasets frequently contain noisy labels, as seen in datasets like ImageNet [9] and Clothing1M [55]. Training deep neural networks with such noisy data would result in poor generalization ability, as these networks can memorize incorrect labels [13, 3].

To mitigate the side effects of label noise, different methods have been proposed [33, 13, 37, 61, 30, 31, 49]. Major existing state-of-the-art methods are based on confident examples selection [61, 2]. Intuitively, those methods first exploit the *memorization effect*, enabling deep neural networks to learn simple patterns shared by the majority of training examples [60, 1, 19]. Since clean labels typically constitute the majority in each noisy class [7, 39], deep neural networks initially fit the training data with correct labels and then progressively fit examples with incorrect labels [13]. To prevent the model from fitting incorrect labels, early stopping is usually employed [44, 3, 50, 4]. Then the *small-loss trick* is used to extract the confident examples with high certainty [21, 35, 28, 38]. If extracted examples have high quality, the performance of a classifier can be enhanced.

---

*Corresponding author: Tongliang Liu (tongliang.liu@sydney.edu.au).

37th Conference on Neural Information Processing Systems (NeurIPS 2023).

To improve the performance of a classifier, it is crucial to ensure that the confident examples have a distribution similar to that of clean data [36, 32]. Specifically, it is necessary to extract not only confident examples that are far from the decision boundary but those that are close to it. The former examples are easy to be identified and extracted. However, the latter examples, which are close to the decision boundary, often become entangled with mislabeled examples, making them challenging to be identified or extracted. In this paper, we discover that on image datasets, *the entanglement of unhelpful style factors with useful content factors* is a key reason that leads to the hard examples becoming hard to be classified, and thus these examples are close to the decision boundary.

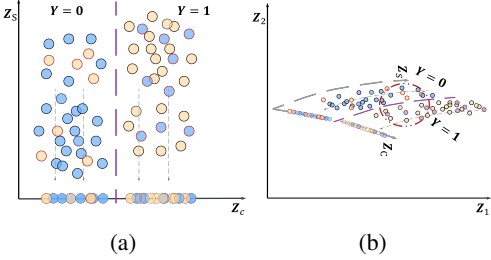

(a)                  (b)

Figure 1: Illustrating the entanglement of content and style factors. The circles on the left side of the dotted line represent examples with the underlying clean label $Y = 0$, while the ones on the right side represent examples with clean label $Y = 1$. The blue filling corresponds to the noisy label $\tilde{Y} = 0$, and the yellow filling corresponds to the noisy label $\tilde{Y} = 1$. Black outlines indicate that the labels of examples are correct, whereas red outlines indicate that the labels of examples are incorrect. In (a), we visualize the underlying style factor $Z_s$ and the underlying content factor $Z_c$. (b) shows the impact of a nonlinear transformation on $Z_s$ and $Z_c$, leading to their entanglement in the new space defined by representations $Z_1$ and $Z_2$.

In Fig. 1, we provide an intuitive understanding of the relationship between hard examples and the entanglement of style factor $Z_s$ and content factor $Z_c$. Let us first assume the underlying content factor $Z_c$ and style factor $Z_s$ are given. In Fig. 1a, by visualizing $Z_s$ and $Z_c$, we can observe a separation between the examples belonging to underlying clean classes $0$ and $1$. In real-world scenarios, however, $Z_c$ and $Z_s$ are not directly available. Instead, existing methods [13, 28] could learn representations $Z_1$ and $Z_2$ from noisy data to extract confident examples. These learned representations, $Z_1$ and $Z_2$, are nonlinear transformations of style factor $Z_s$ and content factor $Z_c$. They do not ensure the disentanglement of $Z_s$ and $Z_c$ [24, 20, 59]. As the example illustrated in Fig. 1b, both $Z_1$ and $Z_2$ contain the information of the style factor $Z_s$ and the content factor $Z_c$. In the illustrated representation space, examples within the region encircled by red dashed lines are close together. The confident examples in this region are entangled with mislabeled examples, making them challenging to be extracted by existing methods without isolating (disentangling) style and content information in learned representations.

We have discussed that the entanglement of style factor $Z_s$ and content factor $Z_c$ in the representation space can create hard examples. However, if we can isolate content from styles, then by focusing solely on the content factor $Z_c$, these hard examples become more discriminative. For example, let's consider projecting the examples onto the $Z_c$-axis by ignoring style factor $Z_s$ in Fig. 1a. The data points on the $Z_c$-axis in Fig. 1b illustrate that even after the nonlinear transformation, the examples remain distinguishable. This motivates us to tackle the problem of inferring content factors for extracting hard examples when learning with noisy labels.

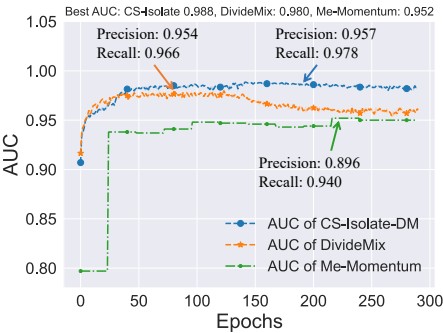

Figure 2: Illustrate the comparison of the performance of confident examples selection of our method CS-Isolate with DivideMix and Me-Momentum on CI-FAR10N Worst [48]. The highest points on AUCs are indicated by arrows. High precision indicates that most of the confident examples are correctly labeled, while recall indicates the fraction of correctly labeled examples out of all examples that are correctly labeled.

To isolate style and content information in learned representations, it is crucial to learn invariant representations of all examples as their styles change. However, when data contains label noise, we can not access clean classes, making it challenging to achieve this isolation. To address this issue, we introduce a method built upon identifiable Variational Autoencoder (iVAE) [23]. The core intuition behind our approach is to construct different styles for each example through various data augmentation techniques. We then encourage both the original and the augmented examples to possess different style factors but maintain

identical content factors, which is a self-supervised learning manner. This initial step could achieve a preliminary level of isolation between style and content factors. As the training process progresses, we can identify and select certain confident examples based on the content factors. Leveraging these confident examples, we further enhance the isolation process by encouraging all examples sharing the same label to share identical content factors. By isolating content and style factors, our method, CS-Isolate, effectively helps existing sample selection methods to extract hard examples typically overlooked by existing sample selection methods. In Fig 2. we evaluate the performance of confident examples selection on CIFAR10N with the noise type "worst" [48] by using two metrics: precision and recall. When comparing these two metrics, CS-Isolate consistently outperforms DivideMix [28] and Me-Momentum [2]. This result demonstrates that our method can not only select confident examples more accurately, as shown by the higher precision, but also capture a larger portion of the correctly labeled examples from the entire dataset, as evidenced by the higher recall. By improving the quality of confident examples with the isolation of the content and style factors, our method effectively helps improve the test accuracy of existing methods. We have also theoretically analyzed the identifiability of content and style factors in Appendix A.

## 2   Background and Related Work

**Problem setup.**     Let's denote $\tilde{D}$ as the distribution of a noisy example $(X, \tilde{Y})$ from the set $\mathcal{X} \times \{1, 2, \ldots, C\}$, where $X$ denotes the variable of instances, $\tilde{Y}$ represents the variable of noisy labels, $\mathcal{X}$ is the feature space, $\{1, 2, \ldots, C\}$ is the label space, and $C$ is the total number of classes. In the learning scenario with noisy labels, clean labels are not observed. Given a noisy training sample $\tilde{S} = \{x_i, \tilde{y}_i\}_{i=1}^{N}$, independently drawn from $\tilde{D}$, the objective is to leverage this sample $\tilde{S}$ to learn a classifier robust against label noise.

**Sample selection methods for learning with noisy labels.**     In learning with noisy labels, major current state-of-the-art (SOTA) methodologies predominantly involve sample selection strategies. These strategies seek to divide the dataset into confident and unconfident examples. The basis of these strategies is the exploitation of the *memorization effect* inherent in deep neural networks. The memorization effect enables the networks to initially grasp and learn the simple patterns, and then learn the complex patterns gradually [60, 1]. Given that clean labels usually form the majority within each noisy class [7, 39], these networks would initially fit the examples with accurate labels, and subsequently fit the examples with incorrect labels over time [13].

Preventing the learning model from fitting incorrect labels is crucial to ensure sample selection quality. To achieve this, strategies such as early stopping are often employed [44, 3]. Additionally, the *small-loss trick* is utilized to identify and extract confident examples with high certainty [28, 38, 51, 18]. Some variation has also been proposed, *e.g.*, some methods opt to reweight examples, thereby decreasing the contribution of mislabeled samples to the overall loss [41, 13]. To guarantee the statistical consistency of algorithms, Jiacheng *et al.* [7] introduces active learning to acquire the labels of randomly chosen examples from unconfident examples to mitigate the bias introduced by sample selection.

Moreover, the set of confident examples often undergoes dynamic changes during the training stage. This is achieved by leveraging semi-supervised learning methods to relabel training instances and reselect confident and unconfident examples using the small loss trick. Various techniques for this purpose have been proposed and empirically demonstrated superior performance, including consistency regularization [27] adopted by [10], MixMatch [5] used by [28], co-regularization by [47], and contrastive learning by [30, 29, 45, 8, 11, 57, 61]. Self-training and co-training have been used by [2] and [35, 13, 21], respectively.

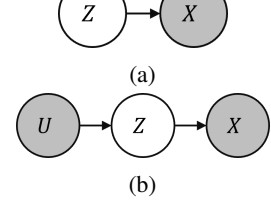

Figure 3: The different data generative processes with or without auxiliary variables.

**Representation learning by generative model.**     Consider a data generation in Fig. 3a, $X$ is the observed data, and $Z$ is the unknown underlying representation to generate $X$. Variational autoencoder framework [24] can be used to learn the latent representation. In this process, a standard normal distribution is utilized as a prior for the latent variables, and a variational posterior $q(Z|X)$ is employed to approximate the unknown underlying posterior $p(Z|X)$. Disentangled representation is very important, which

can allow a rich class of properties to be imposed on the learned representation, such as sparsity and clustering. To disentangle the representation, the variational autoencoder framework has been further expanded by modifying the original loss function, resulting in various algorithms. $\beta$-VAE [16] suggests an adaptation framework that adjusts the weight of the KL term to balance the independence of disentangled factors and reconstruction performance. $\beta$-TCVAE [6] further analyzes the KL term of $\beta$-VAE [16] and only adjusts the total correlation term to achieve disentanglement. HOOD [17] uses the clean labels and domain labels to disentangle content and style factors, but the clean labels and domain labels are unknown in the setting of learning with noisy labels. These methods are toward the goal of disentanglement but do not have theoretical guarantees of identifiability of their inferred latent representations.

**Disentangled latent factors via auxiliary variable.** Recent studies [20, 23] show that theoretical guarantees of identifiability can be achieved if an auxiliary variable related to the representation can be obtained. Khemakhem *et al.* [23] provide identifiability of iVAE with additional inputs by employing the theory of nonlinear Independent Component Analysis (nonlinear ICA) [20]. Intuitively, given a factorized prior distribution over the latent variables that are conditional on an additional auxiliary variable $U$, *i.e.*, the class label and the time index in a time series, the latent factors are identifiable up to a certain degree. The data generation processing is changed to Fig. 3b. To disentangle the representation, the method assumes that each factor of the representation is independent.

# 3 Content and Style Isolation When Learning with Noisy Labels

In this section, we present CS-Isolate, a method designed to help select hard confident examples by content and style isolation.

## 3.1 Preliminaries

**Noisy data generative process.** To learn latent factors by leveraging generative models, the generative model has to model the noisy data generative process. Firstly, we introduce the noisy data generative process. We denote observed variables with gray circles and latent variables with white circles. Specifically, the content factor $Z_c$ is generated by the latent label $Y$. The different style domains $U_s$ give rise to the different style factor $Z_s$. Then, the image $X$ is generated by the combined influence of the style factor $Z_s$ and the content factor $Z_c$. Noisy labels $\tilde{Y}$ are then generated based on the image $X$. In general cases, $Z_c$ and $Z_s$ can also have statistical or causal dependencies. We follow existing work that assumes the content factors are unchanged across different styles [46].

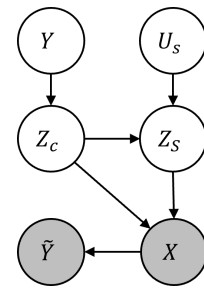

Figure 4: The noisy data generative process.

**Challenge in isolating content from styles without labels.** Isolating content from style without labels is a challenging task [46]. We first introduce the assumptions made by existing methods, as well as difficulties that may be encountered in practice.

Existing methods [26, 23, 46] for isolating content from styles without labels assume that the data augmentation for controlling each style factor can be designed, which means that we can intervene (control) all style factors. For instance, the data augmentations for controlling rotation angle and scaling of images can be designed by using affine transformation [40]. When we apply a data augmentation that rotates an image, the style factor for the rotation angle changes in the augmented image compared to the original. Similarly, when we scale an image, the style factor for scaling becomes different in the augmented image. If we have the data augmentation to control each style factor, then by training a generative model with the augmented images, the model can then identify style factors by comparing the changes in styles between the original and augmented images.

However, the major challenge lies in the fact that we generally cannot design sufficient data augmentations to control all style factors in an image. For instance, in the CIFAR-10 dataset [25], some pictures with the label "horse" contain a person. In this context, the "person" acts as a style factor. Existing data augmentations cannot control this, as they cannot remove the person from images easily. This simple example illustrates that it is generally impossible to control all style factors through data augmentations. Then the assumption required by existing methods usually is hard to satisfy.

The violation of the assumption leads to the learned representations for content containing style information.

**Challenge in isolating content from style with noisy labels.** We generally cannot design data augmentations that can control all style factors in an image. In this case, clean labels are essential to help isolate content from styles. Intuitively, by comparing the change of images with different clean labels that share the same styles, one can infer content factors used for classification. This achieves isolation of content and style. However, when dealing with noisy data, relying on labels becomes problematic as they contain label errors. Images with the same content factors can have different noisy labels, making it fail to infer content factors used for classification. This situation further complicates the task of isolating content from styles, highlighting the challenges posed by noisy labels.

## 3.2 CS-Isolate for Extracting Hard Example

In this paper, we find that a key reason for some hard examples being close to the decision boundary is the entanglement of style factors with content factors. Intuitively, style factors render the learned representation of certain examples less discriminative, thereby making them close to the decision boundary. If content and style information can be isolated, many hard examples would be easily distinguished by ignoring the style information. This is because we have left only content information in the learned representation, making these representations far from the decision boundary.

Inspired by these findings, we propose CS-Isolate, which aims to isolate content from styles for extracting confident examples. To achieve isolation, we utilize self-supervised learning. Specifically, we ensure that original and augmented images maintain the same content factors despite having different style factors due to data augmentations. This achieves a preliminary level of isolating content from styles. As previously mentioned, it is generally impossible to manipulate all style factors through data augmentation. Some style factors remain uncontrolled and stay contained in the learned representations for content. To further encourage the isolation, labels of confident examples are used. Specifically, as training progresses, we identify and select confident examples whose labels are likely to be accurate. By harnessing the labels from these confident examples, we encourage the examples with the same label to have the consistent content factors, irrespective of stylistic variations. This further isolates content from uncontrolled styles. By leveraging different data augmentations and confident examples, our method can effectively separate the learned latent representations into two parts which exclusively contain either style or content information. Subsequently, classifier heads can be trained purely on the representations containing content information. This approach not only improves the identification of hard confident examples but also enhances classification performance.

**Isolating content from styles using auxiliary variables.** We follow existing work using the variational autoencoder with auxiliary variables [23] to isolate content from styles. Let $U_s$ and $U_c$ denote the auxiliary variables that control the style factor $Z_s$ and content factor $Z_c$, respectively. Specifically, by reconstructing an image with the supervision of auxiliary variables $U_s$ and $U_c$, we encourage the images with the same content information but different style information to have the same content factor $Z_c$ but the different style factor $Z_s$. This allows us to isolate content from styles within an image. Then only employing $Z_c$ for selecting hard confident examples.

However, given only noisy data, we face the challenge of the unavailability of auxiliary variables $U_c$ and $U_s$. In response, we devise surrogates for these auxiliary variables. To find a surrogate for the style auxiliary variable $U_s$, we employ different data augmentation techniques, each of which generates a unique style domain with distinct style factors. We then assign the style domain ID as the style auxiliary variable $U_s$. To find a surrogate for the content auxiliary variable $U_c$, we adopt the philosophy of self-supervised learning to assign a unique content ID as the content auxiliary variable $U_c$ to each image. The images that have the same content ID are encouraged to have consistent content factors. Moreover, during the learning process, some confident examples can be extracted. To effectively leverage these examples in learning content factors, we reassign the content ID of confident examples belonging to the same class to be the same. This allows our model to learn consistent content factors for the images in the same class.

**Constructing style auxiliary variables via data augmentations.** For the learning of distinct content factors and style factors, it is essential to construct images from domains with different styles

by using data augmentations. Each image carries its corresponding domain ID as the style auxiliary variable $U_s$. Let $\mathcal{A} = \{A_0, A_1, A_2, \ldots, A_M\}$ represents the set of data augmentations, and $M$ is the number of data augmentations. Note that, for convenience, we use $A_0$ to denote a data augmentation such that applying the data augmentation $A_0$ to the image $x$ does not change it at all, $i.e.$, $x = A_0(x)$.

When applying these diverse data augmentations to an $x$, different augmented images are obtained with different style factors. For the augmented image $x^{A_i}$ obtained by using the $i$-th data augmentations that come from the $i$-th style domain, we can assign its style ID to be $i$. Specifically, for each image $x$, each data augmentation $A_i \in \mathcal{A}$ is applied, resulting in a set containing pairs of the augmented image and the corresponding style ID, $i.e.$,

$$\{(x^{A_i} := A_i(x), U_s^{(x^{A_i})} := i) \mid \forall i \in \{0, ..., M\}\},$$

where $U_s^{x^{A_i}}$ is defined to be the style ID (which serves the style auxiliary variable) of the image $x$ by applying $i$-th data augmentation. In this manner, for each image $x$, we can generate a set of augmented images ($\{x^{A_i}\}_{i=0}^M$), including the original images and the ones with different style factors controlled by different style IDs.

**Constructing content auxiliary variables in a self-supervised manner.** To guide the generative model in learning the consistent content factor across images sharing the same content information, we aim to assign the same content ID (which serves as the content auxiliary variable) to images with the same content information. However, clean labels cannot be obtained in learning with noisy labels, meaning we cannot know which images have the same content information. To construct content auxiliary variables without clean labels, we adopt self-supervised learning. Specifically, we consider that different data augmentations applied to an image typically do not alter its content information. Hence, we can assign a unique content ID to each original image and its augmented versions. This assignment can be mathematically expressed as follows:

$$\mathcal{S}_{all} = \{(x_j^{A_i} := A_i(x), U_s^{(x_j^{A_i})} := i, U_c^{(x_j^{A_i})} := C + j) \mid \forall i \in \{0, ..., M\}, \forall j \in \{1, ..., T\}\},$$

where $T$ is the total number of distinct original images in the training data, $U_c^{(x_j^{A_i})}$ represents the content ID of the $j$-th image in the training data after applying the data augmentation $A_i$, and $C$ is the number of classes.

Moreover, this initial assignment of content IDs can be refined by using confident examples, denoted as $\mathcal{S}_l$. As the training process progresses, we can extract these confident examples by leveraging the small-loss trick [13, 3, 28]. The content IDs are then further refined based on the labels of the examples within the confident examples. The refinement process can be described as:

$$U_c^{(x_j^{A_i})} := y_j^c \mid \forall (x_j, y_j^c) \in \mathcal{S}_l, \forall i \in \{0, ..., M\},$$

where $y_j^c$ is the label of the confident example $(x_j, y_j^c)$. After this refinement process, confident examples in the same class will have the same content ID. Consequently, these refined content IDs could enable the model to learn consistent content factors for the images in the same class.

**Encouraging style and content isolation for extracting hard example.** After obtaining the auxiliary variable, we isolate content from styles by leveraging iVAE [20]. Specifically, the prior distribution of the content factor $Z_c$ is conditional on the auxiliary variable $U_c$, $i.e.$ $P_{\theta_c}(Z_c|U_c)$. Similarly, the prior distribution of the style factor $Z_s$ is conditional on the auxiliary variable $U_s$, $i.e.$ $P_{\theta_s}(Z_s|U_s)$. The $\theta_c$ and $\theta_s$ are the learnable parameters of the distribution. The objective is to maximize the data likelihood which is as follows.

$$\mathbb{E}_{q_D}[p_\theta(X|U_c, U_s)] = \mathbb{E}_{q_D}\left[\int_{z_c, z_s} p_\theta(X|z_c, z_s)p_{\theta_c}(z_c|U_c)p_{\theta_s}(z_s|U_s)\mathrm{d}z_s\mathrm{d}z_c\right], \tag{1}$$

where we use $q_D$ to denote the empirical distribution of the training sample $\mathcal{S}_{all}$. We use the variational inference method to approximate the underlying posterior distribution $p_\theta(Z_c, Z_s|X, U_c, U_s)$ [23, 24]. Specifically, two inference models (encoders) $q_{\phi_c}(Z_c|X)$ and $q_{\phi_s}(Z_s|X)$ are introduced to infer latent variables $Z_c$ and $Z_s$ respectively and model the distribution $q(Z_c, Z_s|X)$ that is used to approximate the distribution $p_\theta(Z_c, Z_s|X, U_c, U_s)$. Therefore, the distribution $q(Z_c, Z_s|X)$ can be decomposed as follows:

$$q_\phi(Z_c, Z_s|X) = q_{\phi_c}(Z_c|X)q_{\phi_s}(Z_s|X).$$

We learn the parameters $\{\theta_c, \theta_s, \phi_c, \phi_c\}$ by maximizing the evidence lower-bound ELBO for each example $(x, u_c, u_s)$. The corresponding loss is:

$$\min_{\phi_c, \phi_s, \theta} \mathcal{L}_{ELBO} := \min_{\phi_c, \phi_s, \theta} \mathbb{E}_{q_D} \left[ -\mathbb{E}_{(z_c, z_s) \sim q_{\phi_c, \phi_s}(Z_c, Z_s | x)} [\log p_\theta(x | z_c, z_s) \right.$$
$$\left. - KL(q_{\phi_c}(z_c | x) || p_{\theta_c}(z_c | u_c)) - KL(q_{\phi_s}(z_s | x) || p_{\theta_s}(z_s | u_s))] \right],$$

where $KL$ is the Kullback–Leibler divergence. Intuitively, when the style changes, the content remains constant. The model has to infer the unchanged content factors for image reconstruction. Thus, the content factors can be isolated from style factors. The content factors can be identified up to the block-identifiable defined in [46].

**Utilizing content information to extract hard examples.** After minimizing the ELBO loss $\mathcal{L}_{ELBO}$, we have learned representations $\hat{Z}_c$ and $\hat{Z}_s$ which exclusively contain either style or content information. Then we employ existing sample-selection-based methods solely focus on $\hat{Z}_c$ for confident example selection, a classification head $f_\psi : \mathcal{Z}_c \to \Delta_{C-1}$ is introduced. The classification head maps the space of content factors to a $C-1$ probability simplex, where $C$ represents the number of classes.

In practice, we propose an *end-to-end* approach to learn the classification head $f_\psi$ and learn to infer content factor for extracting hard confident examples. This approach simultaneously minimizes the evidence lower bound loss $\mathcal{L}_{ELBO}$ and the loss of existing sample-selection-based methods by leveraging the Lagrangian method [22, 12]. The content IDs are refined dynamically during the learning process. The overall loss function $\mathcal{L}_{all}$ is thus given as:

$$\min_{\phi_c, \phi_s, \theta, \psi} \mathcal{L}_{all} = \min_{\phi_c, \phi_s, \theta, \psi} [\mathcal{L}_{ssl} + \lambda_{ELBO} \mathcal{L}_{ELBO} + \lambda_{ref} \mathcal{L}_{ref}],$$

where $\mathcal{L}_{ssl}$ denotes the loss of the sample selection method, which optimizes the parameters of the classification head $\psi$ and the content encoder $\hat{\phi}_c$. The loss $\mathcal{L}_{ref}$ is the cross-entropy loss on confident examples. The hyper-parameter $\lambda_{ELBO}$ and $\lambda_{ref}$ are used to control strength of $\mathcal{L}_{ELBO}$ and $\mathcal{L}_{ref}$, respectively. Here, we illustrate a concrete example of employing the inference model $q_{\hat{\phi}_c}$ in conjunction with DivideMix [28] in an end-to-end implementation. For a detailed walkthrough, please refer to the pseudo-code provided in Appendix B and the original paper [28].

DivideMix [28] applies a Gaussian mixture model to enhance MixMatch [43] for confident example selection and classifiers training. In DivideMix, during each epoch of training, the data is divided into a set of confident examples $\mathcal{S}_l$ and a set of unconfident examples $\mathcal{S}_u$. The confident examples contain the sharpened soft labels mixed by their labels in datasets and predicted labels given by the classification head $f_\psi$. The unconfident examples contain the sharpened soft labels predicted by the classification head $f_\psi$. Then, the semi-supervised learning approach, MixMatch [5] is employed by transforming confident ($\mathcal{S}_l$) and unconfident ($\mathcal{S}_u$) samples into augmented confident ($\mathcal{S}_l'$) and unconfident ($\mathcal{S}_u'$) samples by a linearly mixing.

The overall loss function, composed of a confident sample loss, an unconfident sample loss, and a regularization term, ELBO loss and a classification loss, *i.e.*,

$$\mathcal{L}_{all} = \underbrace{\mathcal{L}_{\mathcal{S}_l} + \lambda_u \mathcal{L}_{\mathcal{S}_u} + \lambda_r \mathcal{L}_{\text{reg}}}_{\text{DivideMix loss}} + \lambda_{ELBO} \mathcal{L}_{ELBO} + \lambda_{ref} \mathcal{L}_{ref}.$$

Intuitively, $\mathcal{L}_{\mathcal{S}_l}$ is a cross-entropy loss for the labeled examples; $\mathcal{L}_{\mathcal{S}_u}$ is the mean squared error for the unlabeled samples; $\mathcal{L}_{\text{reg}}$ is a regularization term to prevent the model from predicting all samples to belong to a single class. These three terms are defined as follows specifically.

$$\mathcal{L}_{\mathcal{S}_l} = -\frac{1}{|\mathcal{S}_l'|} \sum_{x, y^s \in \mathcal{S}_l'} \sum_i y_i^s \log(f_\psi \circ q_{\phi_c}(x)),$$

$$\mathcal{L}_{\mathcal{S}_u} = \frac{1}{|\mathcal{S}_u'|} \sum_{x, y^s \in \mathcal{S}_u'} \|y^s - f_\psi \circ q_{\phi_c}(x)\|_2^2,$$

$$\mathcal{L}_{\text{reg}} = \sum_i \frac{1}{C} \log(1 \Big/ \frac{C}{|\mathcal{S}_l'| + |\mathcal{S}_u'|} \sum_{x \in \mathcal{S}_l' + \mathcal{S}_u'} f_\psi \circ q_{\phi_c}^i(x)),$$

where $y^s$ is the sharpened soft label and $q_{\phi_c}^i$ denotes the $i$-th coordinate of its output on input $x$. In Appendix B, we illustrate our method with other sample selection methods.

Table 1: Precision of confident examples on CIFAR-10N.

|  | Worst | Aggregate | Random 1 | Random 2 | Random 3 |
|---|---|---|---|---|---|
| Me-Momentum | $0.881 \pm 0.011$ | $0.972 \pm 0.002$ | $0.962 \pm 0.005$ | $0.960 \pm 0.003$ | $0.948 \pm 0.005$ |
| DivideMix | $0.951 \pm 0.003$ | $\mathbf{0.989 \pm 0.000}$ | $0.983 \pm 0.000$ | $0.983 \pm 0.000$ | $\mathbf{0.983 \pm 0.000}$ |
| CS-Isolate-DM | $\mathbf{0.956 \pm 0.001}$ | $\mathbf{0.989 \pm 0.000}$ | $\mathbf{0.984 \pm 0.000}$ | $\mathbf{0.985 \pm 0.000}$ | $\mathbf{0.983 \pm 0.000}$ |

Table 2: Recall of confident examples on CIFAR-10N.

|  | Worst | Aggregate | Random 1 | Random 2 | Random 3 |
|---|---|---|---|---|---|
| Me-Momentum | $0.920 \pm 0.026$ | $0.969 \pm 0.019$ | $0.946 \pm 0.021$ | $0.958 \pm 0.022$ | $0.962 \pm 0.018$ |
| DivideMix | $0.966 \pm 0.000$ | $0.963 \pm 0.000$ | $0.975 \pm 0.000$ | $0.976 \pm 0.000$ | $0.977 \pm 0.000$ |
| CS-Isolate-DM | $\mathbf{0.980 \pm 0.000}$ | $\mathbf{0.975 \pm 0.002}$ | $\mathbf{0.980 \pm 0.001}$ | $\mathbf{0.982 \pm 0.001}$ | $\mathbf{0.982 \pm 0.001}$ |

Table 3: Means and standard deviations (percentage) of classification accuracy on FashionMNIST, CIFAR-10 and CIFAR-100.

|  | Fashion-MNIST | | CIFAR-10 | | CIFAR-100 |
|---|---|---|---|---|---|
|  | IDN-0.2 | IDN-0.4 | IDN-0.2 | IDN-0.4 | IDN-0.4 |
| CE | $88.54 \pm 0.32$ | $84.22 \pm 0.35$ | $75.81 \pm 0.26$ | $62.45 \pm 0.86$ | $21.45 \pm 0.70$ |
| Co-Teaching | $91.21 \pm 0.31$ | $89.10 \pm 0.29$ | $80.96 \pm 0.31$ | $73.41 \pm 0.78$ | $28.04 \pm 1.43$ |
| Forward | $90.05 \pm 0.43$ | $86.27 \pm 0.48$ | $74.64 \pm 0.26$ | $60.21 \pm 0.75$ | $26.75 \pm 0.93$ |
| T-Revision | $91.58 \pm 0.31$ | $89.46 \pm 0.42$ | $76.15 \pm 0.37$ | $65.09 \pm 0.37$ | $27.23 \pm 1.13$ |
| BLTM | $91.20 \pm 0.27$ | $82.42 \pm 1.51$ | $77.50 \pm 1.30$ | $63.20 \pm 4.52$ | $35.67 \pm 1.97$ |
| CausalNL | $90.84 \pm 0.31$ | $90.01 \pm 0.45$ | $80.91 \pm 1.14$ | $79.08 \pm 0.50$ | $34.02 \pm 0.95$ |
| Me-Momentum | $92.85 \pm 0.64$ | $90.06 \pm 0.51$ | $90.86 \pm 0.21$ | $86.66 \pm 0.91$ | $58.38 \pm 1.28$ |
| DivideMix | $94.85 \pm 0.15$ | $92.28 \pm 0.13$ | $94.93 \pm 0.15$ | $94.16 \pm 0.35$ | $70.50 \pm 0.25$ |
| CS-Isolate-DM | $\mathbf{95.16 \pm 0.07}$ | $\mathbf{94.40 \pm 0.09}$ | $\mathbf{95.90 \pm 0.10}$ | $\mathbf{95.54 \pm 0.06}$ | $\mathbf{73.11 \pm 0.36}$ |

Table 4: Means and standard deviations (percentage) of classification accuracy on CIFAR-10N.

|  | Worst | Aggregate | Random 1 | Random 2 | Random 3 |
|---|---|---|---|---|---|
| CE | $77.69 \pm 1.55$ | $87.77 \pm 0.38$ | $85.02 \pm 0.65$ | $86.14 \pm 0.24$ | $86.12 \pm 0.16$ |
| Co-Teaching | $82.04 \pm 0.06$ | $91.11 \pm 0.10$ | $89.61 \pm 0.18$ | $88.98 \pm 0.11$ | $89.49 \pm 0.06$ |
| Forward | $79.79 \pm 0.46$ | $88.24 \pm 0.22$ | $86.88 \pm 0.50$ | $86.14 \pm 0.24$ | $87.04 \pm 0.35$ |
| T-Revision | $80.48 \pm 1.20$ | $88.52 \pm 0.17$ | $88.33 \pm 0.32$ | $87.71 \pm 1.02$ | $87.79 \pm 0.67$ |
| BLTM | $68.21 \pm 1.67$ | $79.41 \pm 1.00$ | $78.09 \pm 1.03$ | $76.99 \pm 1.23$ | $76.26 \pm 0.71$ |
| CausalNL | $82.41 \pm 0.24$ | $90.43 \pm 0.14$ | $89.03 \pm 0.02$ | $89.06 \pm 0.05$ | $89.21 \pm 0.13$ |
| Me-Momentum | $84.21 \pm 0.70$ | $91.34 \pm 0.16$ | $89.51 \pm 0.42$ | $90.14 \pm 0.28$ | $89.62 \pm 0.31$ |
| DivideMix | $92.48 \pm 0.16$ | $94.11 \pm 0.21$ | $94.77 \pm 0.15$ | $94.79 \pm 0.14$ | $94.79 \pm 0.15$ |
| CS-Isolate-DM | $\mathbf{94.28 \pm 0.07}$ | $\mathbf{95.34 \pm 0.10}$ | $\mathbf{95.36 \pm 0.14}$ | $\mathbf{95.34 \pm 0.12}$ | $\mathbf{95.48 \pm 0.14}$ |

## 4 Experiments

In this section, we introduce the setting of our experiments and compare our experimental results with existing methods. Most of our experiments are left out in Appendix C due to the limited space.

### 4.1 Experiment Setting

**Dataset and noise type.** We evaluate our methods on three synthetic noise datasets FashionMNIST [54], CIFAR-10 [25], and CIFAR-100 [25], and two real-world label-noise datasets, CIFAR-10N [48] and Clothing1M [55]. FashionMNIST includes 70,000 images of size $24 \times 24$, categorized into 10 classes with 60,000 for training and 10,000 for testing. Both CIFAR-10 and CIFAR-100 contain 50,000 training images and 10,000 test images; CIFAR-10 comprises 10 classes, while CIFAR-100 includes 100 classes. The image size in both CIFAR datasets is $32 \times 32 \times 3$. To generate noisy labels for these clean datasets, we employ the instance-dependent noisy label generation methods proposed in [52]. CIFAR-10N, a noisy version of CIFAR-10, includes five types of label noise: "worst", "aggregate", "random 1", "random 2", and "random 3", all annotated by humans. The noise rates are 40.21%, 9.03%, 17.23%, 18.12%, and 17.64%, respectively. Clothing1M contains 1 million images with real-world noisy labels for training and 10,000 images with clean labels for testing.

Table 5: Means and standard deviations (percentage) of classification accuracy on Clothing1M.

| CE | Decoupling [35] | MentorNet [21] | Co-teaching [13] | Forward [39] |
|---|---|---|---|---|
| 68.88 | 54.53 | 56.79 | 60.15 | 69.91 |

| T-Revision [53] | BLTM-V [56] | CausalNL [58] | DivideMix [28] | CS-Isolate-DM |
|---|---|---|---|---|
| 70.97 | 73.39 | 72.24 | 74.76 | **74.92** |

**Network structure and optimization.** We implemented our method using PyTorch 1.12.1 and performed experiments on the NVIDIA Tesla V100. For fair comparisons, we chose PreAct ResNet-18 [15] as the backbone of the encoder $q_{\phi_c}$ for Fashion-MNIST, CIFAR-10, CIFAR-100, and CIFAR-10N, a ResNet-50 [14] with ImageNet pre-trained weight for Clothing1M. We employed SGD with a momentum of 0.9 and a weight decay of 0.0005 to optimize the encoder $q_{\phi_c}$ and the classifier head $f_\psi$. We used Adam with default parameters to optimize the encoder $q_{\phi_s}$ and the decoder $p_\theta$. For experiments on synthesis datasets, the initial learning rate for SGD was set at 0.02 and for Adam at 0.001. The batch size is 64. For experiments on Fashion-MNIST, our network was trained for 100 epochs. Both learning rates were reduced by a factor of 10 after 80 epochs. For experiments on CIFAR-10, CIFAR-100, and CIFAR-10N, our network was trained for 300 epochs. Both learning rates were reduced by a factor of 10 after 150 epochs. For experiments on Clothing1M, our network was trained for 80 epochs with a batch size of 32. The initial learning rate for SGD was set at 0.002 and for Adam at 0.001. Both learning rates were reduced by a factor of 10 after 40 epochs. The encoder $q_{\phi_c}$ and the classifier head $f_\psi$ were warmed up on noisy data for 10 epochs for CIFAR-10 and CIFAR-10N, warmed up for 30 epochs for CIFAR-100, 5 epochs for Fashion-MNIST, and 1 epoch for Clothing1M. The dimensions of $Z_c$ and $Z_s$ were set at 32. The data augmentation techniques include shift scale rotation, random crop and horizontal flip, random brightness contrast, color jitter, and random to gray. For all experiments, we set $M = 1000$, $\lambda_r = 1$ and $\lambda_{ELBO} = 1e-3$. $\lambda_{ref}$ was increased gradually then kept constant at $1e-3$ after 140 epochs. For synthetic datasets, we set $\lambda_u$ as 0 and 15 for the noise rates of 0.2 and 0.4 in FashionMNIST and CIFAR-10 datasets and as 100 in CIFAR-100. For CIFAR-10N, we set $\lambda_u$ as 50 in the "worst" noise type and 0 in the rest of the noise types. For Clothing1M, we set $\lambda_u$ as 0.

**Baselines and measurements.** We compare our method against several state-of-the-art techniques: (1) CE: the standard cross-entropy loss, with the model trained directly on noisy data; (2) Co-Teaching [13]: involves training two networks and uses the small-loss trick to select confident examples for each other; (3) Forward [39]: applies a transition matrix to correct the loss function; (4) T-Revision [53]: revises the transition matrix to enhance performance; (5) Me-momentum [2]: uses confident examples to refine the classifier, which is then used to update confident examples alternately; (6) BLTM [56]: uses a Bayes optimal label to estimate the transition matrix; (7) CausalNL [58]: employs a causal model to assist the learning of the classifier; (8) DivideMix [28]: employs two networks to select confident examples and uses semi-supervised techniques to utilize unlabeled examples. We report the precision, recall and AUC for the confident examples selection. We also report the test accuracy on the test dataset. The value is the average over the last 10 epochs.

## 4.2 Confident Examples Quality

To evaluate the quality of the confident examples, we examine the precision and recall of these examples. The precision quantifies the proportion of accurately labeled examples within the set of confident examples. On the other hand, recall measures the proportion of accurately labeled examples in the confident examples relative to the total number of correct labels. The experimental results are displayed in Tab. 1 and Tab. 2. The results indicate that the precision performance of our proposed method matches the state-of-the-art levels. However, the recall of our method surpasses that of existing methods. This demonstrates that our method can identify more confident examples without compromising precision. This suggests that our approach effectively maintains high precision and improves recall, thus selecting more confident examples with correct labels.

## 4.3 Classification Accuracy

We conducted extensive experiments on three synthetic noise datasets (Fashion-MNIST, CIFAR-10 and CIFAR-100) and two real-world datasets (CIFAR-10N and Clothing1M). For the synthetic

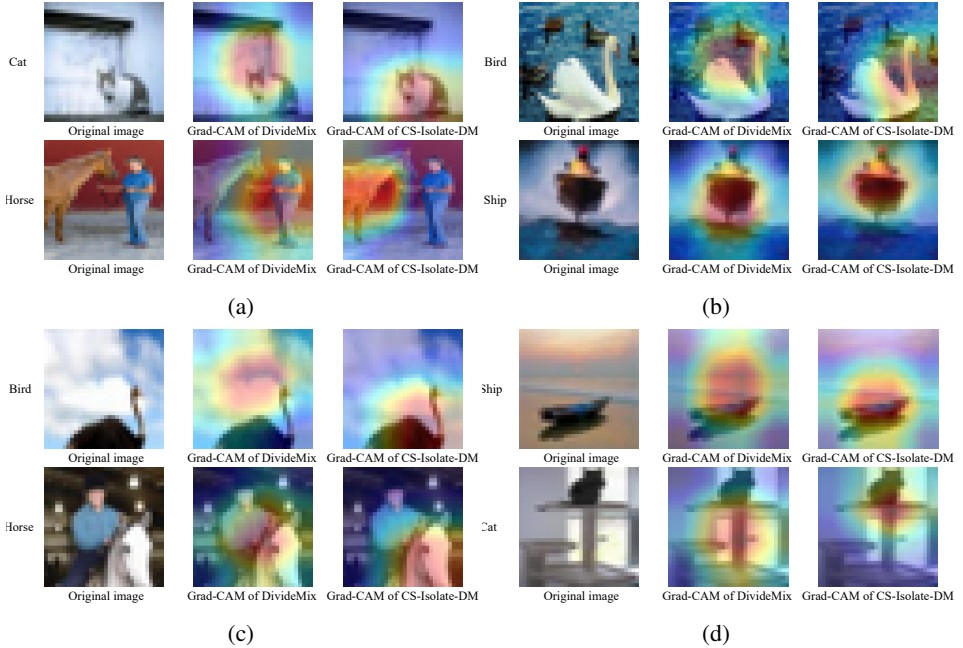

Figure 5: Grad-CAM visualizations of hard confident examples. CS-Isoalte-DM successfully identifies these confident examples, but DivideMix does not. The activation map of CS-Isolate-DM predominantly highlights semantic objects, whereas DivideMix emphasizes non-object pixels.

datasets, we employed instance-dependent noisy label generation methods, as proposed by [52]. We experimented with noise rates of 0.2 and 0.4, denoted by IDN-0.2 and IDN-0.4 respectively. The experiment results are presented in Tab. 3, Tab. 4 and Tab. 5. Our proposed method outperforms existing methods in terms of test accuracy on both synthetic and real-world datasets containing label noise. Notably, as the noise rate increases, the performance gap between our method, CS-Isolate-DM, and the existing methods becomes more pronounced. This highlights the robustness and effectiveness of our approach in scenarios with higher levels of label noise.

## 4.4 Hard Confident Examples Visualization

The proposed method is expected to select confident examples based on content factors rather than style factors. To analyze this, we use Grad-CAM [42] to visualize the regions used to select confident examples. The visualization results are shown in Fig. 5. The experiment is conducted on the real-world dataset CIFAR-10N, and the noise type is "worst". The experiment results demonstrate that our method can correctly focus on the object in images. Specifically, when there exist uncontrolled style factors, *e.g.*, the person near the horse, the activation maps for CS-Isolate-DM can successfully focus on the right object used for classifying the horse instead of uncontrolled style factors. In contrast, the baseline method, DivideMix, focuses on the style factors that are not related to the class "horse" and fails to select these confident examples.

## 5 Conclusion

This paper is motivated by the fact that only focusing on content factors such as semantic information makes examples more discriminative. We, therefore, proposed a novel CS-Isolate approach to infer and isolate the content information for classification. This is achieved by leveraging variational inference and constructing auxiliary variables via data augmentation techniques to modify style factors while regularizing content factors using confident examples. By training existing sample-selection-based methods with our inferred content factors, CS-Isolate improves their effectiveness in learning hard examples and classification accuracy on different image datasets.

## Acknowledgments and Disclosure of Funding

Tongliang Liu is partially supported by the following Australian Research Council projects: FT220100318, DP220102121, LP220100527, LP220200949, and IC190100031. Dong Xu is partially supported by the Hong Kong Jockey Club Charities Trust under Grant 2022-0174, the Hong Kong Research Grant Council under General Research Fund (17203023), the Startup Funding and the Seed Funding for Basic Research for New Staff from The University of Hong Kong, and the funding from UBTECH Robotics.

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

## A A Theoretical View of Content and Style Isolation When Learning with Noisy Labels

**Noisy data generative process.** Recall the data generative process in our main paper, to learn latent factors by leveraging generative models. The generative model has to model the data generative process of noisy data. Firstly, we introduce the generative process of noisy data. We denote observed variables with gray color and latent variables with white color. Specifically, the content factor $Z_c$ is generated by the latent label $Y$. The different style domain $U_s$ give rise to the different style factor $Z_s$. Subsequently, the image $X$ is generated by the combined influence of the style factor $Z_s$ and the content factor $Z_c$. Noisy labels $\tilde{Y}$ are then generated based on the image $X$. In general cases, $Z_c$ and $Z_s$ can also have statistical or causal dependencies. We follow existing work that assumes the content factors are invariant across different styles [46].

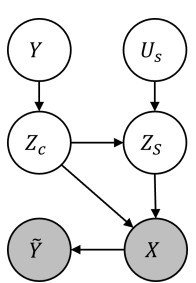

Figure 6: The noise data generative process.

Firstly, we introduce the concept of an *uncontrolled style factor*. This refers to a specific style factor, denoted $Z_{s'}$, that remains invariant when a data augmentation $A$ is applied. To formalize this concept, consider an invertible function $f : \mathcal{Z} \times \mathcal{X}$. Let $\mathcal{A}$ denote a set of data augmentations, where each augmentation $A$ is a subset ranging from 1 to $M$. Additionally, let $P(A)$ represent a probability distribution over the set of these augmentations $\mathcal{A}$. Now, consider $Z_{s'}$ as a subset of style factors drawn from a larger set $Z_s$. We partition the latent factors $z$ of each instance $x$ into three distinct parts: uncontrolled style factor $Z_{s'}$, content factor $Z_c$, and style factors influenced by data augmentation $Z_{s/s'}$, such that $f^{-1}(x) = [z_{s'}, z_c, z_{s/s'}] = z$. The term $z_{s/s'}$ denotes the set of style factors with style factors $z_{s'}$ excluded.

**Definition 1 (Uncontrolled Style Factors)** *We say that a style factor $Z_{s'}$ as uncontrolled under the following conditions:*

*For any augmentation $A \sim P(A)$, and for any instance $x$, the first $n_{s'}$ components of the inverse function $f^{-1}(x)$ remain unchanged even when $A(x)$ is applied, i.e., $f^{-1}(x)_{1:n_{s'}} = f^{-1}(A(x))_{1:n_{s'}}$.*

Here, $f^{-1}(x)_{1:n_{s'}}$ is defined as the underlying partition that contains only and all the information related to the style factor $Z_{s'}$ of the instance $x$.

**Why do confident examples encourage content-style isolation?** Here, we explain the reason that confident examples encourage content-style isolation. Suppose that there exist some uncontrolled style factors that cannot be adjusted or manipulated through data augmentation. This implies that for an image $x$, these style factors remain unaffected, regardless of the data augmentation techniques employed. For instance, as discussed in our paper, in the CIFAR-10 dataset [25], some pictures with the label "horse" contain a person. In this context, the "person" acts as a style factor. Existing data augmentations cannot control this, as they cannot remove the person from images easily.

It's essential to understand that although data augmentation cannot control all style factors, it still offers the benefit of "partial isolation". If we consider a situation where the uncontrolled style factors are not effects of other style factors, then we can isolate other style factors affected by data augmentation from content factors after matching the data likelihood [34]. This result is a modified application of *block-identifiability* [46].

Specifically, let $\hat{Z}_{c,s'} := \hat{Z}_{1:n_c+n_{s'}}$ be a partition of learned representations of a generative model, where $\hat{Z}_{c,s'}$ contain *all* and *only* information about $Z_c$ and $Z_{s'}$. Suppose that assumptions of Theorem 4.2 in [34] are satisfied, $\hat{Z}_{c,s'}$ is guaranteed to be learned, thereby allowing for the partial isolation of the remaining style factors, denoted as $Z_{s/s'}$.

Despite $\hat{Z}_{c,s'}$ is guaranteed to be learned, the information from the uncontrolled style factor $Z_{s'}$ is entangled with the content factor $Z_c$ in the learned $\hat{Z}_{c,s'}$. To isolate this information further, the employment of *confident examples* is necessary.

Specifically, we can generate $\hat{Z}_{c,s'}$ using an invertible function according to the label $\hat{Y}$ of the confident example, and subsequently reconstruct image $x$. Following the data likelihood matching [34], the information related to the uncontrolled style factors becomes apparent due to the establishment of a

---

**Algorithm 1** CS-Isolate-DM

---

**Input:** A noisy dataset $\tilde{\mathcal{S}}$, a style ID list $U_s$, a data augmentation set $\mathcal{A}$, confident threshold $\tau$, total epoch $T_{max}$ .

1: $f_{\psi_1}, f_{\psi_2}, q_{\phi_c^1}(Z_c|X), q_{\phi_c^2}(Z_c|X) \leftarrow \text{WarmUP(S)}$;
2: **For** T $= 1, \ldots, T_{max}$:
3:    $\mathcal{X}, \mathcal{U} \leftarrow \text{Co} - \text{guessing}(\tilde{\mathcal{S}}, f_{\psi_1}, f_{\psi_2}, q_{\phi_c^1}, q_{\phi_c^2}, \tau)$;
4:    $\mathcal{X}', \mathcal{U}' \leftarrow \text{MixUp}(\mathcal{X}, \mathcal{U})$;
5:    **For** k=1, 2:
6:       Sample $(x, \tilde{y}, u_c) \sim \tilde{\mathcal{S}}, u_s \sim U_s$;
7:       $\tilde{x} \leftarrow A_{u_s}(x)$;
8:       Feed $\tilde{x}$ to encoders $\hat{q}_{\phi_c^k}$ and $\hat{q}_{\phi_s^k}$ to content factors $z_c$ and style factors $z_s$, respectively;
9:       Feed $u_c$ and $u_s$ to decoders $\hat{p}_{\theta_c^k}$ and $\hat{p}_{\theta_s^k}$ to get the prior $\hat{p}_{\theta_c^k}(Z_c|u_c)$ and $\hat{p}_{\theta_s^k}(Z_s|u_s)$;
10:      Feed $z_c$ and $z_s$ to decoders $\hat{p}_{\theta^k}$ to get the reconstruct image $\hat{\tilde{x}}$;
11:      Feed $z_c$ to classifier head $f_{\psi_k}$ to predicted label $\hat{y}$;
12:      Calculate the loss using Eq. 2 and update networks;
   **Output:** The inference networks and classifier heads $q_{\phi_c^1}, q_{\phi_c^2}, f_{\psi_1}, f_{\psi_2}$.

---

---

**Algorithm 2** CS-Isolate-Co

---

**Input:** A noisy dataset $\tilde{\mathcal{S}}$, a style ID list $U_s$, a data augmentation set $\mathcal{A}$, Total epoch $T_{max}$ .

1: **For** T $= 1, \ldots, T_{max}$:
2:    **Fetch** mini-batch $\bar{S}$ from $\tilde{\mathcal{S}}$;
3:    **Obtain** $\bar{S}_1 = \arg\min_{S':|S'| \geq R(T)|\bar{S}|} \ell(f_{\psi_1}, q_{\phi_c^1}, S')$;
4:    **Obtain** $\bar{S}_2 = \arg\min_{S':|S'| \geq R(T)|\bar{S}|} \ell(f_{\psi_2}, q_{\phi_c^2}, S')$;
5:    **For** k=1, 2:
6:       Sample $(x, \tilde{y}, u_c) \sim \bar{S}_k, u_s \sim U_s$;
7:       $\tilde{x} \leftarrow a_{u_s}(x)$;
8:       Feed $\tilde{x}$ to encoders $\hat{q}_{\phi_c^k}$ and $\hat{q}_{\phi_s^k}$ to content factors $z_c$ and style factors $z_s$, respectively;
9:       Feed $u_c$ and $u_s$ to decoders $\hat{p}_{\theta_c^k}$ and $\hat{p}_{\theta_s^k}$ to get the prior $\hat{p}_{\theta_c^k}(Z_c|u_c)$ and $\hat{p}_{\theta_s^k}(Z_s|u_s)$;
10:      Feed $z_c$ and $z_s$ to decoders $\hat{p}_{\theta^k}$ to get the reconstruct image $\hat{\tilde{x}}$;
11:      Feed $z_c$ and $z_s$ to classification model $f_{\psi_k}$ to predicted labels $\hat{y}$;
12:      Calculate the loss using Eq. 3 and update networks;
   **Output:** The inference networks and classifier heads $q_{\phi_c^1}, q_{\phi_c^2}, f_{\psi_1}, f_{\psi_2}$.

---

one-to-one mapping between the label $\hat{Y}$ and $\hat{Z}_{c,s'}$. This consequently forces examples with identical labels to share the same $\hat{Z}_{c,s'}$, regardless of any alterations in the uncontrolled style factor $Z'_s$. This approach, therefore, ensures that styles changes don't affect the derived content representation for the same label.

It is worth mentioning that to fully isolate uncontrolled style factors and content factors, it requires that there exists confident examples with all possible uncontrolled style factors. This can be hard to achieve when learning with noisy labels. Therefore, in general, the selected confident examples can only encourage isolation but can not fully isolate uncontrolled style factors and content factors.

# B   Apply CS-Isolate to Existing Methods for Learning with Noisy Labels

**Applying CS-Isolate to DivideMix.**     DivideMix [28] uses two classifiers to select confident examples for each other. To utilize the unlabeled data, they combine the semi-supervised technique MixMatch [5]. Specifically, the classifiers, after warmed up, are used to calculate the loss of examples. They use a Gaussian Mixture Model (GMM) to divide the examples into confident and unlabeled examples. Finally, confident and unlabeled examples are used to train the models based on the MixMatch algorithm. Our method can be plugged into DivideMix easily. Specifically, we use a decoder $q_\phi(Z_c, Z_s|X)$ to obtain content factor $Z_c$ and style factor $Z_s$. A classifier head $f_\psi$ is used to

---

**Algorithm 3** CS-Isolate-Me

---

**Input:** A noisy training dataset $\tilde{\mathcal{S}}$, a noisy validation dataset $\tilde{\mathcal{S}}_v$, a style ID list $U_s$, a data augmentation set $\mathcal{A}$, iteration number $T_{inner}, T_{outer}$ .

1: **Initialize** encoders ($\hat{q}_{\phi_c^0}$ and $\hat{q}_{\phi_s^0}$), decoders ($\hat{p}_{\theta_c^0}$, $\hat{p}_{\theta_s^0}$ and $\hat{p}_{\theta^0}$), a classification model $f_{\psi_0}$ by using the noisy training data and early stopping;
2: **For** i = 1, ..., $T_{outer}$:
3:   **For** j = 1, ..., $T_{inner}$:
4:     **Update** the extracted confident examples $\mathcal{S}_l$ by using $\hat{q}_{\phi_c^{j-1}}$ and $f_{\psi_{j-1}}$;
5:     **Train** networks by using the loss in Eq. 4 on confident examples $\mathcal{S}_l$;
6:     **Obtain** $\hat{q}_{\phi_c^j}$ and $f_{\psi_j}$ through the highest noisy validation accuracy throughout the training procedure;
7:     **Break** and output $\hat{q}_{\phi_c^{j-1}}$ and $f_{\psi_{j-1}}$ if the highest validation accuracy is non-increasing in the loop;
8:   **Re-initialize** encoders ($\hat{q}_{\phi_c^0}$ and $\hat{q}_{\phi_s^0}$), decoders ($\hat{p}_{\theta_c^0}$, $\hat{p}_{\theta_s^0}$ and $\hat{p}_{\theta^0}$), a classification model $f_{\psi_0}$;
9:   **Train** networks by using the loss in Eq. 4 on confident examples $\mathcal{S}_l$;
10:   **Obtain** $\hat{q}_{\phi_c^0}$ and $f_{\psi_0}$ through the highest noisy validation accuracy throughout the training procedure;
11:   **Break** and output $\hat{q}_{\phi_c^{j-1}}$ and $f_{\psi_{j-1}}$ if the highest validation accuracy is non-increasing in the loop;

**Output:** The inference network and the classification model $q_{\phi_c}, f_\psi$.

---

predict labels, and only the content factors $Z_c$ are used as input. The prior distribution of the content factors $Z_c$ is conditional on the auxiliary variable $U_c$, *i.e.*, $P_{\theta_c}(Z_c|U_c)$, where $U_c$ is the content ID. Similarly, the prior distribution of the style factor $Z_s$ is conditional on the auxiliary variable $U_s$, *i.e.*, $P_{\theta_s}(Z_s|U_s)$, where $U_s$ is the style ID. A decoder $P_\theta(X|Z_c, Z_s)$ is used to reconstruct input images. The combination of CS-Isolate and DivideMix is called CS-Isolate-DM. The loss function of CS-Isolate-DM is shown in Eq. 2. Algorithm 1 delineates the full algorithm.

$$\mathcal{L}_{dm} = \underbrace{\mathcal{L}_{\mathcal{S}_l} + \lambda_u \mathcal{L}_{\mathcal{S}_u} + \lambda_r \mathcal{L}_{\text{reg}}}_{\text{DivideMix loss}} + \lambda_{ELBO}\mathcal{L}_{ELBO} + \lambda_{ref}\mathcal{L}_{ref}. \tag{2}$$

**Applying CS-Isolate to Co-Teaching.** Co-Teaching [13] uses two classifiers to select confident examples for each other. Proposed CS-Isolate can be embedded in Co-Teaching easily. Similar to CS-Isolate-DM, we use a decoder $q_\phi(Z_c, Z_s|X)$ to obtain content factors $Z_c$ and style factors $Z_s$. A classifier head $f_\psi$ is used to predict labels, and only the content factors $Z_c$ are used as input. The prior distribution of the content factors $Z_c$ is conditional on the auxiliary variable $U_c$, *i.e.*, $P_{\theta_c}(Z_c|U_c)$, where $U_c$ is the content ID. Similarly, the prior distribution of the style factor $Z_s$ is conditional on the auxiliary variable $U_s$, *i.e.*, $P_{\theta_s}(Z_s|U_s)$, where $U_s$ is the style ID. A decoder $P_\theta(X|Z_c, Z_s)$ is used to reconstruct input images. We call the combined method as CS-Isolate-Co. Let $\mathcal{S}_l$ be the confident examples selected by another classifier head. For each network, the loss is defined as:

$$\mathcal{L}_{co} = \underbrace{\mathbb{E}_{(x,\tilde{y})\sim\tilde{\mathcal{S}}}[\mathbb{1}_{(x,\tilde{y})\in\mathcal{S}_l}\ell_{ce}(f_\psi \circ q_{\phi_c}(x), \tilde{y})]}_{\text{Co-Teaching loss}} + \lambda_{ELBO}\mathcal{L}_{ELBO}, \tag{3}$$

where $\ell_{ce}$ is the cross-entropy loss, $\mathbb{1}$ is the indicator function. The algorithm of CS-Isolate-Co-Teaching is summarized in Algorithm 2.

We use a PreAct ResNet-18 as the backbone. We used Adam with default parameters to optimize the encoder $q_{\phi_c}$, classifier head $f_\psi$, the encoder $q_{\phi_s}$ and the decoder $p_\theta$. The initial learning rate is 0.001, divided by 10 after 80 epochs.

**Applying CS-Isolate to Me-Momentum.** Me-Momentum proposes to use one classifier to select confident examples. Then the parameters of the classifier will be reinitialized, and the classifier will be trained on the confident examples. The confident examples and the parameters of the classifier are updated alternately. We combine the Me-Momentum with our method and call it CS-Isolate-Me. Let

Table 6: Means and standard deviations (percentage) of classification accuracy on CIFAR-10.

|  | Sym-20% | Sym-40% | Sym-60% | Sym-80% | Pair-45% |
|---|---|---|---|---|---|
| CE | $84.12 \pm 0.20$ | $78.82 \pm 0.38$ | $64.80 \pm 0.47$ | $47.39 \pm 0.69$ | $66.09 \pm 0.96$ |
| Co-Teaching | $88.74 \pm 0.19$ | $84.23 \pm 0.81$ | $77.93 \pm 0.75$ | $29.57 \pm 1.39$ | $76.65 \pm 2.97$ |
| Forward | $88.31 \pm 0.23$ | $82.73 \pm 0.47$ | $76.58 \pm 1.59$ | $47.18 \pm 4.63$ | $76.76 \pm 4.94$ |
| T-Revision | $87.83 \pm 0.63$ | $84.46 \pm 0.73$ | $77.39 \pm 0.83$ | $57.61 \pm 3.93$ | $72.81 \pm 7.01$ |
| BLTM | $76.54 \pm 1.37$ | $73.50 \pm 1.38$ | $58.94 \pm 1.84$ | $39.28 \pm 4.00$ | $67.97 \pm 1.45$ |
| CausalNL | $89.68 \pm 0.09$ | $86.37 \pm 0.09$ | $79.54 \pm 0.09$ | $29.72 \pm 0.03$ | $71.53 \pm 0.37$ |
| Me-Momentum | $91.44 \pm 0.33$ | $88.39 \pm 0.34$ | $82.58 \pm 0.30$ | $62.70 \pm 0.59$ | $66.41 \pm 0.17$ |
| DivideMix | $95.93 \pm 0.04$ | $94.51 \pm 0.12$ | $94.55 \pm 0.07$ | $92.43 \pm 0.13$ | $70.86 \pm 0.87$ |
| CS-Isolate-DM | $\mathbf{96.05 \pm 0.13}$ | $\mathbf{95.57 \pm 0.12}$ | $\mathbf{94.65 \pm 0.10}$ | $\mathbf{92.57 \pm 0.10}$ | $\mathbf{87.54 \pm 0.83}$ |

Table 7: Precision ratio (percentage) of confident examples on CIFAR-10N.

|  | Worst | Aggregate | Random 1 | Random 2 | Random 3 |
|---|---|---|---|---|---|
| Co-Teaching | $89.49 \pm 0.11$ | $96.61 \pm 0.04$ | $96.06 \pm 0.03$ | $95.65 \pm 0.02$ | $96.01 \pm 0.03$ |
| Me-Momentum | $88.14 \pm 1.10$ | $97.16 \pm 0.19$ | $96.18 \pm 0.47$ | $95.96 \pm 0.27$ | $94.78 \pm 0.47$ |
| CS-Isolate-Co | $\mathbf{90.88 \pm 0.08}$ | $\mathbf{97.25 \pm 0.04}$ | $\mathbf{96.95 \pm 0.03}$ | $\mathbf{96.60 \pm 0.06}$ | $\mathbf{96.82 \pm 0.05}$ |
| CS-Isolate-Me | $\mathbf{90.74 \pm 1.16}$ | $\mathbf{97.72 \pm 0.07}$ | $\mathbf{96.64 \pm 0.21}$ | $\mathbf{96.92 \pm 0.17}$ | $\mathbf{96.14 \pm 0.22}$ |

Table 8: Recall ratio (percentage) of confident examples on CIFAR-10N.

|  | Worst | Aggregate | Random 1 | Random 2 | Random 3 |
|---|---|---|---|---|---|
| Co-Teaching | $89.11 \pm 0.07$ | $95.43 \pm 0.04$ | $92.59 \pm 0.04$ | $93.19 \pm 0.02$ | $93.00 \pm 0.04$ |
| Me-Momentum | $92.02 \pm 2.63$ | $96.90 \pm 1.91$ | $94.58 \pm 2.07$ | $95.80 \pm 2.20$ | $96.22 \pm 1.82$ |
| CS-Isolate-Co | $\mathbf{90.53 \pm 0.06}$ | $\mathbf{96.10 \pm 0.02}$ | $\mathbf{93.44 \pm 0.04}$ | $\mathbf{94.13 \pm 0.06}$ | $\mathbf{93.79 \pm 0.03}$ |
| CS-Isolate-Me | $\mathbf{95.76 \pm 1.55}$ | $\mathbf{97.76 \pm 0.79}$ | $\mathbf{97.10 \pm 0.45}$ | $\mathbf{96.34 \pm 0.81}$ | $\mathbf{96.70 \pm 0.89}$ |

$\mathcal{S}_l$ be the confident examples selected by the classifier of the last iteration . For each network, the loss is defined as:

$$\mathcal{L}_{me} = \mathbb{E}_{(x,\tilde{y}) \sim \tilde{\mathcal{S}}}[\mathbb{1}_{(x,\tilde{y}) \in \mathcal{S}_l} \ell_{ce}(f_\psi \circ q_{\phi_c}(x), \tilde{y}] + \lambda_{ELBO} \mathcal{L}_{ELBO}. \tag{4}$$

where $\ell_{ce}$ is the cross-entropy loss, $\mathbb{1}$ is the indicator function. The algorithm of CS-Isolate-Me-Momentum is summarized in Algorithm 3.

# C   Experiments

In this section, we first introduce the experiment results of the proposed methods, including the classification of CS-Isolate-DM on class-dependent label noise, sample selection quality of CS-Isolate-Co and CS-Isolate-Me on the real-world dataset CIFAR-10N, and classification performance of CS-Isolate-Co and CS-Isolate-Me on real-world the dataset CIFAR-10N. Second, we provide the details of data augmentation used in methods. Then, we conduct the ablation study for CS-Isolate-DM. Finally, we visualize the easy confident examples, content factors, and style factors.

## C.1   Experiments on Class-dependent label noise

We report the classification performance of CS-Isolate-DM on class-dependent label noise, including symmetry-flipping noise and pair-flipping noise. The dataset used in the experiment is CIFAR-10. The experiment results are shown in Tab. 6. The experiment results demonstrate that CS-Isolate-DM can also perform well under class-dependent label noise.

## C.2   Applying CS-Isolate to Existing Sample-Selection Methods

We combine our method with existing methods including DivideMix, Co-Teaching, and Me-Momentum.

Table 9: Means and standard deviations (percentage) of classification accuracy on CIFAR-10N.

|  | Worst | Aggregate | Random 1 | Random 2 | Random 3 |
|---|---|---|---|---|---|
| Co-Teaching | $82.04 \pm 0.06$ | $91.11 \pm 0.10$ | $89.61 \pm 0.18$ | $88.98 \pm 0.11$ | $89.49 \pm 0.06$ |
| Me-Momentum | $84.21 \pm 0.70$ | $91.34 \pm 0.16$ | $89.51 \pm 0.42$ | $90.14 \pm 0.28$ | $89.62 \pm 0.31$ |
| CS-Isolate-Co | $\mathbf{83.93 \pm 0.17}$ | $\mathbf{91.30 \pm 0.10}$ | $\mathbf{90.57 \pm 0.14}$ | $\mathbf{90.14 \pm 0.03}$ | $\mathbf{90.76 \pm 0.15}$ |
| CS-Isolate-Me | $\mathbf{86.95 \pm 0.13}$ | $\mathbf{91.49 \pm 0.17}$ | $\mathbf{90.30 \pm 0.12}$ | $\mathbf{90.18 \pm 0.07}$ | $\mathbf{90.22 \pm 0.21}$ |

Table 10: Data Augmentation Techniques

| Data augmentation | Description |
|---|---|
| Shift scale rotation | Randomly shifts, scales, and rotates the image. |
| Random crop | Randomly crops the image to a specified height and width. |
| Horizontal flip | Horizontally flips the image randomly. |
| Random brightness contrast | Randomly changes the brightness and contrast of the image. |
| Color jitter | Randomly adjusts image color properties. |
| Random to gray | Converts the image to grayscale with a specified probability. |

In the experiments for Co-Teaching and Me-Momentum, we use a PreAct ResNet-18 as the backbone. We use SGD with momentum 0.9 and weight decay $10^{-4}$ to optimize the encoder $q_{\phi_c}$ and classifier head $f_\psi$. We used Adam with default parameters to optimize the encoder $q_{\phi_s}$ and the decoder $p_\theta$. The network is trained for 100 epochs. The initial learning rate for SGD is 0.01, and for Adam is 0.001. The learning rate is divided by 10 after 40 epochs and 80 epochs.

## C.3 Improves Sample Selection Quality with CS-Isolate

We conducted experiments on CIFAR-10N, a dataset reflecting real-world label noise. We illustrate the precision and recall ratios of our confident examples in Tab. 7 and Tab. 8. By employing our method, existing methods achieve improvements in terms of precision and recall. The experiment results indicate that our approach can efficiently improve the quality and number of confident examples.

## C.4 Comparison of Classification Performance

The test accuracy of the baseline methods, as well as the combination of our proposed methods and the baselines, is shown in Tab. 9. The results demonstrate that improving the quality of the confident examples by using our method boosts the classification performance of the existing methods.

## C.5 Data Augmentation Details

The detailed description of data augmentation techniques used in our method is shown in Tab 10. When generating a data augmentation $A_i \in \mathcal{A}$, the probabilities to apply shift scale rotation, random crop and horizontal flip, random brightness contrast, color jitter, and random to gray are 0.5, 1, 0.5, 0.5, 0.8, and 0.2, respectively, then the implementation details of the data augmentations will be recorded for the replaying. For instance, if a data augmentation $A_i$ flips the image horizontally, its behavior will be recorded. When the data augmentation $A_i$ is used during the training process, the images used to train the network will be applied a horizontal flip.

## C.6 Ablation Study

In this subsection, we present the results of the ablation study on the hyper-parameters $\lambda_{ELBO}$, $\lambda_{ref}$ and the dimensions of $Z_c$ and $Z_s$. The experiments are conducted on the real-world dataset CIFAR-10N with the noise type "worst". The experiment results are shown in Fig. 7. The experiment results show that $\lambda_{ELBO}$ and $\lambda_{ref}$ are not sensitive in the range from 0.0005 to 0.005. In our experiments, we set the value of both $\lambda_{ELBO}$ and $\lambda_{ref}$ as 0.001, which is the middle value between 0.0005 and 0.005. For the ablation study on the dimension of $Z_c$ and $Z_s$, the test accuracy increases gradually until the dimension is 32. After the dimension is larger than 64, the test accuracy decreases. We set the dimension as 32 in our experiments.

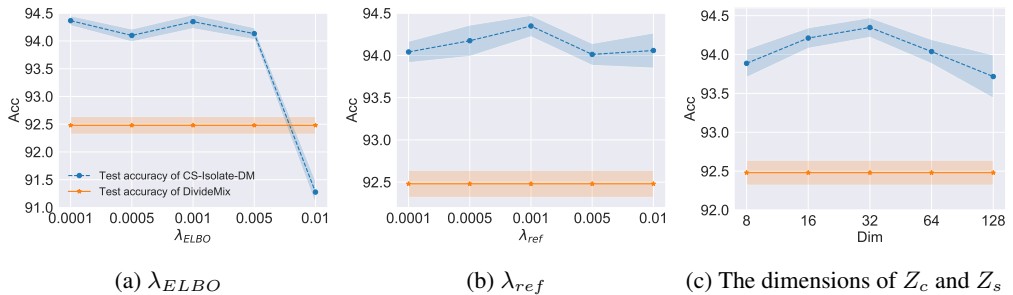

(a) $\lambda_{ELBO}$     (b) $\lambda_{ref}$     (c) The dimensions of $Z_c$ and $Z_s$

Figure 7: Ablation study on the hyper-parameters $\lambda_{ELBO}$, $\lambda_{ref}$ and the dimensions of $Z_c$ and $Z_s$.

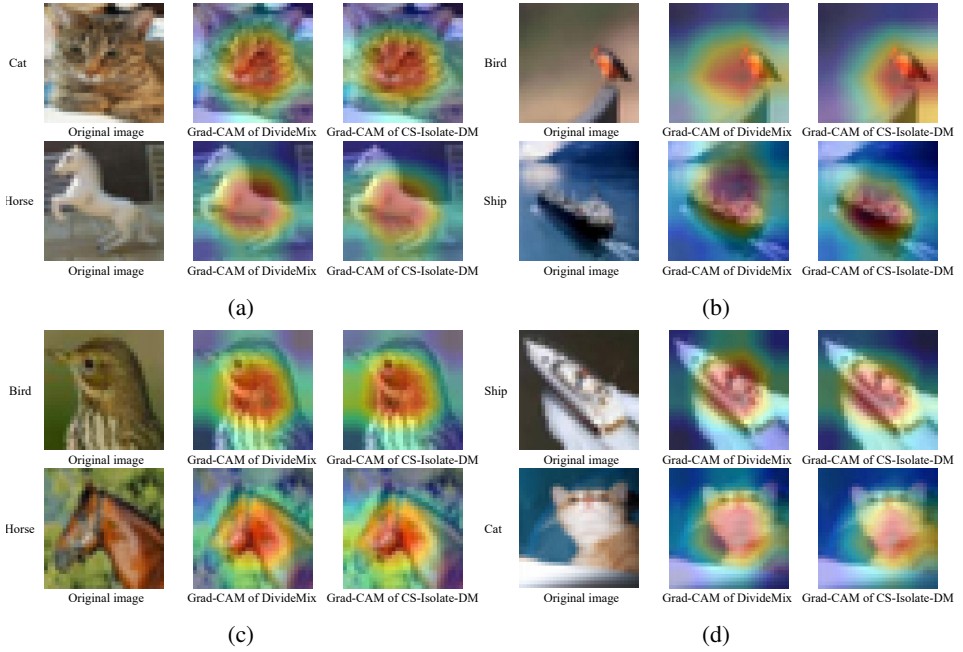

Figure 8: Grad-CAM visualizations of easy confident examples. Both CS-Isoalte-DM and DivideMix successfully identify these confident examples. The activation map of CS-Isolate-DM predominantly highlights semantic objects.

## C.7 Visualization of Grad-CAM on Easy Confident Examples

We visualize the Grad-CAM for CS-Isolate-DM and DivideMix on easy confident examples. The easy confident examples are the examples identified successfully by both CS-Isolate-DM and DivideMix. The dataset is CIFAR-10N, and the noisy type is "worst". Fig. 8 shows the visualizations of easy confident examples. When the confident example is easy, the Grad-CAM visualizations for CS-Isolate-DM and DivideMix do not differ significantly. The activation maps for both CS-Isolate-DM and DivideMix can focus on the objects. However, when the confident example is hard, only the activation maps for CS-Isolate-DM can focus on the objects, which has already been shown in the main paper.

## C.8 Visualization of Grad-CAM for Content Factors and Style Factors

We visualize the Grad-CAM of Style and Content Factors for CS-Isolate-DM. The visualization results are shown in Fig. 9. Grad-CAM of content factors mainly concentrates on the objects, while Grad-CAM of style factors mainly concentrates on other pixels in the images. The visualization results demonstrate that CS-Isolate-DM can isolate content and style factors successfully.

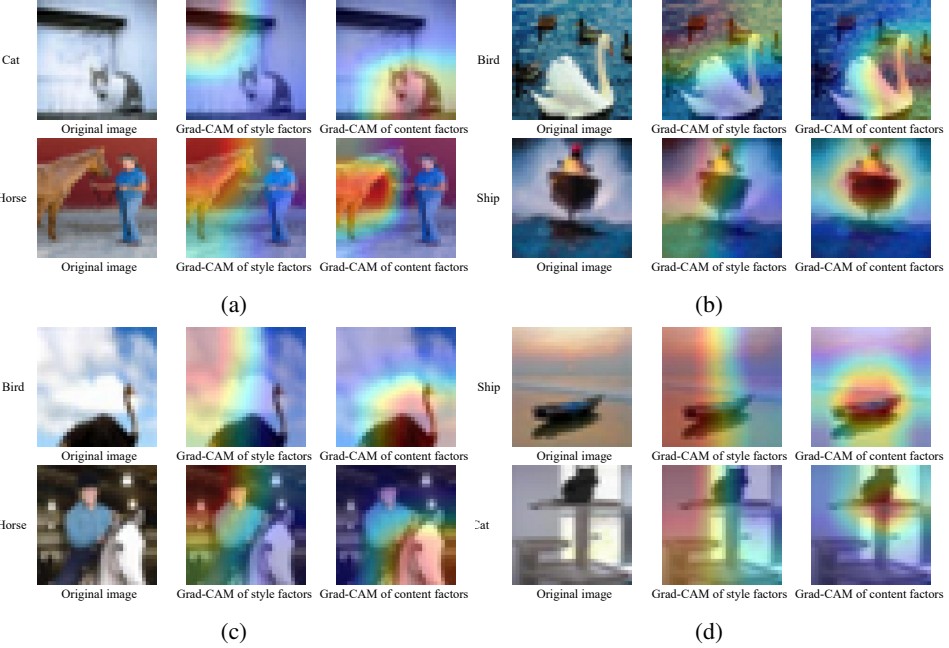

Figure 9: Grad-CAM Visualizations of Style and Content Factors for CS-Isolate-DM. The activation map corresponding to the content factor prominently highlights semantic objects, indicating the model's emphasis on capturing meaningful context. Conversely, the activation maps associated with the style factor predominantly focus on non-object pixels across the images. The results show that CS-Isolate-DM can isolate the content factors from styles.

