# OpenReview forum: "CS-Isolate: Extracting Hard Confident Examples by Content and Style Isolation"
_NeurIPS.cc/2023/Conference — NeurIPS 2023 poster_

### Official Review · Reviewer_25vx · 2023-06-30

**Soundness:** 2 fair
**Presentation:** 3 good
**Contribution:** 2 fair
**Rating:** 3
**Confidence:** 4

**Summary:**

Summary: This paper presents a solution to infer content factors for classification when learning with the noisy labels especially with hard samples that are close to decision boundary. In order to achieve the goal, they tend to separate the style information from the semantic information with some techniques including data augmentations. Some experimental evaluations and comparisons are shown to prove the effectiveness of the paper.

**Strengths:**

The idea of disentangled representation is widely applied for many deep generative models. In this paper, the solution to separate style factors from semantic information also falls into this category which helps to reduce the effect from noisy labels.

**Weaknesses:**

(1)	First of all, it is not clear for the label noise level, what does random 1, 2, 3 indicate in the experimental evaluations for all the Tables 1-4. Because in the regime of robust learning, usually the percentage of noisy labels is an important parameter in the experimental design. Without clearly stating the noise level, it is very difficult for readers to compare if the approach can work well in the high noise ratio level. Nor does this comparison show fairness as random 1 can be a very low level label noise for method A and a high level label noise for method B.
(2)	As the authors mentioned, if the style and semantic information are important to their idea and the design of the method, it is imperative to visualize the style and semantic information extracted from their images and show it is processed by their method. Lacking of visualization make the readers confusing and difficult to understand. Namely, the method is not well justified and stays on the safe ground without analyzing the semantic information (with visualization). Specifically, it is important to show say for MNIST datasets, semantic information implies the numbers and style information indicates the writing style in this case.
(3)	Moreover, there are quite a lot of related work on applying the embedding (dimensionality reduction) to disentangle the latent space for improving the denoising performance (e.g. generative models). After reading the paper, I feel the method claimed is merely a small tweak of the existing methods along this line where the author called it “style and semantic spaces”, while it can actually be more general for all subspaces.
(4)	The papers with similar ideas can be found:
a.	Learning disentangled representations via product manifold projection, ICML 2021.
b.	Learning disentangled representations in the imaging domain, Medical Image Analysis, 2022
Just to name a few, especially in the reference [b], the solution to introduce the disentangled representations is very similar to the paper. Moreover, it is well known that using disentangled representations can boost performance including denoising with noisy labels.
Namely, from the high level, the result shown in this paper is only a subset or a specific case of the previous work, which is not novel enough. While the specific application is somewhat interesting, I donot see it is as a Neurips paper.


**Questions:**

(1) As mentioned before, the overall novelty for this work is small given the fact that disentangled representations can boost the performance including denoising.

(2) The experimental evaluations are not sufficient and questionable due to lack of details. Typically, the level of noisy label percentage is not reported, making the comparison not convincing. Instead, the author simply used some vague statements like "random 1", "random 2" "worst" which leaves many unanswered questions. For instance, how large is the "random 1" noise level ? Is the worst case representing a more than 50% corrupted noisy label space ? What does "Aggregate" mean ? (Is it implies voting from multiple people for labeling).
I tried very hard to find the answers but fail to see the detailed meanings of them.



**Limitations:**

The paper is proposed with limited novelty, questionable experimental design and unclear writing.

---

> ### Author Rebuttal · Authors · 2023-08-10
>
> >**Q1. The authors used confusing words "random 1, 2, 3, worst" to call noise types on CIFAR-10N.**
>
> **A1**. The terms "Random 1", "worst," and "aggregate" are directly from the original paper [1]. They correspond to specific noise types on the real-world noise dataset CIFAR-10N. For example, “Worst" is a real-world noise type with a noise rate of  40.21%.
>
> >**Q2. Visualize style and content factors.**
>
> **A2**. **Visualize style and content factors.** In Fig. 3 of Rebuttal PDF, we visualize Grad-CAM for content factors and style factors extracted by our method, respectively. Grad-CAM of content factors mainly concentrates on the objects, in contrast, Grad-CAM of style factors mainly concentrates on other pixels in the images.
>
>  **Grad-CAM Analysis of DivideMix.** We also employ Grad-CAM to analyze the representation learned by DivideMix, and the results (also shown in Fig. 1 Rebuttal PDF) reveal that the corresponding heatmap highlights non-semantic parts of the images. This implies that DivideMix's learned representation contains style information that doesn't contribute usefully to the classification, causing the examples to lie closer to the decision boundary and then can not be selected as confident examples.
>
> **Comparison with CS-Isolate-DM.**  Moreover, we employ Grad-CAM to analyze content factors learned by CS-Isolate-DM (shown in Fig. 1 Rebuttal PDF). It shows that the content factors concentrate on objects, effectively isolating content from style factors that concentrate on other parts of the images. As a result, these examples are not hard examples anymore and are selected as confident examples by our method.
>
> >**Q3. Noise type Random 1 leads to different noise levels for different methods, then the comparison is unfair.**
>
> **A3**.The comparison is fair. It's important to note that for the same noise type on CIFAR-10N (such as Random 1), the exact same noisy examples are used across all methods being compared. Therefore, the noise level is fixed for all methods.
>
> >**Q4. Applying disentangled representations for learning with noisy labels is a special case and is not novel.**
>
> **A4**. Existing disentangled representation methods can not be directly applied to learning with noise labels. We would like to discuss our method in detail and highlight our novelty. Hopefully, it can address your concern well.
>
>  **Theoretical Alignment.** Our method aligns with the framework that employs auxiliary variables to disentangle latent representations and nonlinear ICA, a proven approach when auxiliary variables are available. However, applying it to noisy data is a novel challenge, as auxiliary variables are not accessible for label-noise datasets.
>
> **New Challenge with Noisy Data.** Given a noisy dataset without access to clean labels, the disentanglement frameworks cannot be employed due to missing auxiliary variable information. Our paper addresses this gap by targeting two new questions: 1) what auxiliary variables need to be constructed, and 2) how to effectively construct them.
>
> **Answers to these questions.** We identify that both style and content auxiliary variables are required to achieve disentanglement through intuitive examples and theoretical analyses in our main paper and Appendix A. Specifically, although existing theoretical work [1] suggests that data augmentation can fully disentangle style, we found that their assumption is too strong. We emphasize that in practice, it's impossible to find sufficient data augmentations to control all style factors. Some factors remain uncontrolled, and their information will be mixed into content factors. We propose that leveraging confident examples is key to further disentangling this information.
> After having identified what auxiliary variables are required, we carefully design CS-Isolate which effectively disentangles content and style factors by properly constructing both style and content auxiliary variables.
>
> **Peer Agreement on Novelty or Insight.** Reviewers NWTZ, MtkC, and fCED have acknowledged the newness, innovation, and novelty of our method. Additionally, as agreed by Reviewers fCED, koUE, and MtkC, our work provides new insight into how to extract hard confidence examples in labeling with noisy labels.
> >**Q5. The solution to introduce the disentangled representations is very similar to the paper [3].**
>
> **A5**. Paper [3] is a survey paper on methods for disentangling representations in the image domain. It of course contains some applications that leverage the framework of using auxiliary variables to disentangle representations when auxiliary variables are available. However, our work goes beyond this by addressing missing auxiliary variables in learning with label noise.
>
> As aforementioned in A4, unlike scenarios where auxiliary variables are available, given only noisy labels, we have to focus on determining what auxiliary variables need to be constructed, and how to effectively construct them. Moreover, we provide theoretical explanations to support the necessity of using confident examples; we also offer new insight into why certain correct examples are hard to be extracted as confident examples.
>
> >**Q6. This paper shares a similar idea with the papers [4].**
>
> **A6**. The ideas must not be similar, as the assumptions for achieving disentanglement are completely different. Specifically, the method in paper [4] relies on having non-i.i.d. examples (data pairs) to learn the disentangled representations. However, our method is in the settings of i.i.d. examples and employs auxiliary variables to disentangle latent representations.
>
> ### References
>
> [1] Learning with Noisy Labels Revisited: A Study Using Real-World Human Annotations, ICLR22.
>
> [2] Self-Supervised Learning with Data Augmentations Provably Isolates Content from Style, NeurIPS21.
>
> [3] Learning disentangled representations in the imaging domain, Medical Image Analysis 22.
>
> [4] Learning disentangled representations via product manifold projection, ICML21.

---

> > ### Comment · Reviewer_25vx · 2023-08-21
> > **About the rebuttal**
> >
> > Thanks for the rebuttal. I appreciate the responses from the author to my question.
> >
> > After reading them, I feel I still have questions to some of the aspects:
> > (1) Fundamentally, as the paper is talking about content and style. I think it is important that the paper is motivating the idea well by showing the visualization at the beginning and then process it to show that the method can really improve. Otherwise, it is too abstract.
> >
> > (2) It is important to show the performance with respect to various noise level. Namely it is important to plot performance curves for the accuracy of the algorithm with varying noise levels. The reason is that random noise can have different variance so that the comparison may not be fair and it is possible that different approaches are subject to different strengths of noise. While the authors respond to this point in the rebuttal. I believe it is not clear enough as this point should be very clear in the main body of the paper. For instance, how is the performance at real world noise ? How is the performance when the noise is 30% ? Without showing curves with varying noise level, it is really difficult to convince the readers. In addition, how the is performance when the noise is asymmetric ?
> >
> > Due to these above issues in the motivation and evaluations, I maintain my original score.

---

> > > ### Author Response · Authors · 2023-08-21
> > >
> > > Thank you for your response! We would like to address your followup questions here. Hopefully, this can address your concerns well.
> > >
> > > > **Followup Q1. I think it is important that the paper is motivating the idea well by showing the visualization at the beginning and then processing it to show that the method can really improve. Otherwise, it is too abstract.**
> > >
> > > Followup A1. We truly appreciate this valuable feedback. This concern should be already solved. We have shown the visualization of the learned content and style in Rebuttal PDF.  The results show that our method can effectively disentangle content and style factors. It also shows that our method can sufficiently identify hard examples that existing methods can not discover. Please also refer to A2 of our rebuttal. We will carefully add the visualization and discussion in the final version.
> > >
> > >
> > > >  **Followup Q2. The main paper does not display performance under various real-world noise levels.**
> > >
> > > Followup A2.  We have displayed the performance of different methods under various real-world noise levels in Table 4 of our main paper. The real-world noise CIFAR10-N dataset is employed.  To the best of our knowledge, CIFAR10-N is the only real-world label-noise dataset containing different known noise levels. The noise levels contain 9.03% (Aggregate), 18.12% (Random 2),  and 40.21% (Worst).  The result shows that our method outperforms other strong baselines.
> > >
> > >
> > >
> > > >  **Followup Q3. Missing results  about noise level 30%**
> > >
> > > Followup A3. the main paper, we presented the performance of different methods for instance-dependent noise on Fashion-MNIST, CIFAR-10, and CIFAR-100 datasets at noise levels of 20% and 40%. In our main paper, we also demonstrated the performance on CIFAR10-N containing real-world label noise at levels of 9.03%, 18.12%, and 40.21%.
> > > Our Rebuttal PDF also includes the performance of different methods on  CIFAR-10 at noise levels of 20%, 40%, 60%, and 80% for symmetric noise.
> > >
> > > Given this extensive evaluation across different noise types around 20% and 40%, the additional experiments at the 30% noise level may not be important.
> > >
> > >
> > > > **Followup Q4. How is the performance when the noise is asymmetric?**
> > >
> > > Followup A4. In Table 3 of our Rebuttal PDF,  the pairflip noise at a 45% level (termed Pair 45%) is asymmetric noise. It's worth noting that the instance-dependent noise used in our main paper is inherently asymmetric. The results are presented in Table 3 of the main paper on different datasets (Fashion-MNIST, CIFAR-10, and CIFAR-100). In these evaluations, our method consistently outperformed other leading benchmarks.
> > >
> > > Many thanks,
> > >
> > > Authors

---

### Official Review · Reviewer_MtkC · 2023-07-03

**Soundness:** 3 good
**Presentation:** 3 good
**Contribution:** 3 good
**Rating:** 7
**Confidence:** 3

**Summary:**

This paper provides insights into the extraction of hard and confident examples from noisy datasets, which is an important problem when learning with noisy labels. The authors discover that by removing irrelevant style information from the data, hard examples can become more discernible. Building on this finding, they design an innovative method that leverages existing identifiable results of latent variables. With no side information available with learning with noisy labels, this method constructs the necessary information by using data augmentations and confident examples for the identification of latent variables. They support their solution with theoretical analysis, demonstrating its empirical performance on different datasets.

**Strengths:**

This paper studies an important problem in machine learning. Specifically, the problem that how to extract hard confident examples when dealing with datasets containing label noise. The ``easy’’ confident examples have been well studied and can be easily identified by using the different techniques such as small-loss trick and memorization effect. However, how to identify the hard and confident examples have only been discussed by a few work. How to effectively identify the hard and confident examples still remains unclear. This problem is important as the hard confident examples usually plays an important role for shaping the decision boundary and improve the classification performance, which mentioned in the previous paper.

The authors present new insight about hard confident examples. They claim that some examples become difficult to discriminate due to the entanglement of non-useful style information in the learned representation. This finding is interesting and easy to understand, has been surprisingly overlooked in existing work. This finding may pave a new way on discover hard examples in noisy datasets.

The provided solution is innovative for me. To enjoy the identifiable of latent factors by using variational inference, the availability of side information is necessary. Nonetheless, when learning in the presence of label noise, this side information is not readily accessible. To navigate this challenge, the authors ingeniously devise supervised information for style and content factors by utilizing different data augmentations and confident examples. They also provide intuitive reasoning and a theoretical analysis to demonstrate how identifiability can be achieved, reinforcing the originality and rigor of their approach.


**Weaknesses:**

The reproducibility of the proposed method could be enhanced. Specifically, the authors mentioned employing various data augmentation methods such as shift scale rotate, random crop and horizontal flip, random brightness contrast, color jitter, and random grayscale. However, it remains unclear how concretely the different augmentations are sampled for each dataset, which could affect replication efforts.

The paper's experimental setup could benefit from further expansion. For synthetic noise datasets, only instance-dependent label noise is used. The performance under widely used noise types like symmetric noise and pairflip noise remains untested. Additionally, an analysis of the method's sensitivity in response to different dimensions of latent variables Z could provide a more comprehensive understanding of the model's robustness.

The last two paragraphs of introduction are redundant and could be effectively condensed into one concise paragraph. Additionally, it might be beneficial to move the identifiable analysis from the appendix to the main paper after introducing the method for a more seamless reader experience.

The paragraph ``Disentangled latent factors via auxiliary variable’’ feels somewhat abstract. As these techniques are leveraged to encourage isolation, further explanation could help reduce the reading difficulty for readers. Providing more in-depth discussion in appendix might make this critical point more accessible.


**Questions:**

1) If the pseudo-code for CS-Isolate-Me is missing or omitted, it would indeed be helpful to understand its differences compared to CS-Isolate-Co. Both methods appear to use a very similar loss function.
2) How the evidence Lower Bound (ELBO) are derived, it is neither the main paper nor the appendix.
3) What will be the results of the proposed method on CIFAR10 and CIFAR100 with symmetric noise and pairflip noise.
4) Does the proposed method sensitive to the dimensionality of $z$?
5) Could the authors provide more concrete details about the application of various augmentations on each dataset?


**Limitations:**

The authors have mentioned a limitation in appendix that the assumption to achieve the fully isolation of style and content variables is strong. Specifically, achieving the fully isolation of uncontrolled style factors and content factors demands the existence of confident examples of all possible uncontrolled style factors. Consequently, the authors' methods typically encourage, but do not guarantee, the full isolation of these uncontrolled style and content factors.
Nevertheless, I believe it's crucial to include a section dedicated to limitations in the main paper and move this point to that section. More broadly, a method's limitations can encompass a range of aspects, such as computational requirements, sensitivity to hyperparameters, and generalizability across different datasets or noise types, among other factors. I recommend the authors carefully review the proposed method to identify if any of these limitations.

---

> ### Author Rebuttal · Authors · 2023-08-10
>
> >**Q1. It remains unclear how concretely the different augmentations are sampled for each dataset.**
>
> **A1**. In our paper, we utilize 6 types of data augmentation techniques that follow [1].
>
> **Detailed types of data augmentation.** These techniques include shift scale rotation, random crop and horizontal flip, random brightness contrast, color jitter, and random to gray.
> **How to generate style ID.** We generate different styles via the combination of different types of data augmentation. Each type of data augmentation has a probability to decide whether this augmentation will be applied. The probability of the random crop is 1; The probability of shift scale rotation, horizontal flip, random brightness contrast and random to gray is 0.5; The probability of color jitter is 0.8. Each combination acts as a style and will be assigned a unique style ID.
> We set the number of different data augmentation $M$ to be 200 in experiments.
>
> >**Q2. Noise types like symmetric noise and pairflip noise remain untested.**
>
> **A2**. We have conducted experiments on the symmetric noise and pairflip noise on CIFAR-10. The noise rate in symmetric noise is 0.2, 0.4 0.6 and 0.8. The noise rate in pairflip is 0.45. The experiment settings are the same as the experiment in the paper. The results are shown in the Tab. 3 of Rebuttal PDF. CS-Isolate-DM performs better than DivideMix.
>
> >**Q3. Sensitivity in response to different dimensions of latent variables Z.**
>
> **A3**. We have conducted the ablation study on the dimension of $Z_c$ and $Z_s$ on CIFAR-10N, and the noise type is ”Worst”. The experiment settings are as same as the description in Sec. 4.1 of our paper.
> The value of the dimension of $Z_c$ and $Z_s$ is 4, 8, 16, 32, 64, 128 and 256 respectively. The experiment results are shown in Tab. 2 of Rebuttal PDF. The test accuracy increases gradually until the dimension is 64. After the dimension is larger than 64, the test accuracy decreases slightly.
>
>
> >**Q4. The paragraph “Disentangled latent factors via auxiliary variable” feels somewhat abstract.**
>
> **A4**. Thanks for your advice. A brief explanation is shown as follows. We will include more details in the final version.
>
> **Understanding Latent Factors.**
> Latent factors represent underlying attributes or features in the data. For example, in an image of a car, latent factors might include the color, shape, and size of the car. These factors might be mixed or entangled with each other, making it challenging to manipulate or understand them individually.
>
> **Introducing Auxiliary Variables.**
> Auxiliary variables can be thought of as additional information or constraints that assist in the disentanglement process. These could be external labels, time indexes in a time series, or other side information that correlates with the latent factors.
>
> **Model Design.**
>
> Typically, a model such as a Variational Autoencoder (VAE) may be employed to learn a representation of the data. The model would have specific components that are designed to capture the content, style, or other latent factors.
>
> **Incorporating Auxiliary Variables.**
> If the auxiliary variables are given, these can be used as conditional inputs to the model, guiding the learning process; or can be used to define additional loss functions or regularization terms that encourage the model to learn disentangled representations; or to ensure that specific neurons are responsive to particular factors, thus forcing a separation of these factors in the learned representation.
>
> >**Q5. Combine two paragraphs of introduction. Pseudo-code for CS-Isolate-Me**
>
> **A5**. Thanks. We will carefully merge two paragraphs and move the theoretical analysis into the main paper as suggested. We will also add the pseudo-code of CS-Isolate-Me in the final version.
>
> >**Q6. How the evidence Lower Bound (ELBO) is derived.**
>
> **A6**. We apologize for any confusion surrounding the derivation of the Evidence Lower Bound (ELBO) in our work. Our ELBO is derived by directly applying Bayesian inferences similar to ELBO in iVAE framework [2]. The idea is Instead of using a combined latent variable $z$, we partition it into two components: content factors $z_c$ and style factors $z_s$. This leads to the formulation of two separate KL-divergence terms, $KL(q(z_c|x)||p(z_c|u_c))$ and $KL(q(z_s|x)||p(z_s|u_s))$, rather than the aggregated term $KL(q(z|x)||p(z|u))$ found in Eq. (53) of paper [2]. This separation allows us to disentangle content and style factors and aligns with our problem setting.
>
>
> ### References
>
> [1] Self-Supervised Learning with Data Augmentations Provably Isolates Content from Style, NeurIPS21.
>
> [2] Variational Autoencoders and Nonlinear ICA: A Unifying Framework, AISTATS20.

---

### Official Review · Reviewer_koUE · 2023-07-07

**Soundness:** 3 good
**Presentation:** 2 fair
**Contribution:** 3 good
**Rating:** 5
**Confidence:** 2

**Summary:**

This paper focuses on improving the ability to extract hard confident examples (close to the decision boundary), a critical aspect of noisy label learning.  The authors find that a key reason for some hard examples being close to the decision boundary is due to the entanglement of style factors with content factors. In response to this finding, they strive to maintain the consistency of content factors across all examples within the same underlying clean class, even as their style information undergoes changes. This approach is grounded in the belief that the content factors should remain invariant, thereby allowing for a more accurate extraction of hard confident examples. Their proposed method, CS-Isolate, embodies this concept and leverages various data augmentation to alter the styles while regularizing content factors based on some confident examples. CS-Isolate demonstrates robust performance when tested on synthetic noise datasets.

**Strengths:**

* The paper exhibits a commendable level of organization, facilitating a smooth and intuitive reading experience. Its logical structure aids in the comprehension of complex concepts and methodologies.
* The inclusion of visual aids for the toy example, significantly enhances the reader's understanding. These illustrations provide tangible insights into the underlying intuition of the proposed method, making the abstract concepts more accessible.
* The proposed method achieves new SOTA performance on various benchmark datasets.

**Weaknesses:**

* The authors base their proposed method on the premise that hard examples being close to the decision boundary is due to the entanglement of unhelpful style factors with useful content factors. However, this key finding appears to lack empirical support from experimental results, which weakens the foundation of their argument.

* I still can’t wrap my head around how the proposed method solves this challenge of identifying or designing data augmentations that can manipulate all style factors in an image.

* The proposed method seems to focus on learning style factors based on artificial styles introduced through data augmentations, rather than modelling the natural style of an image. If the authors' assertion that the proximity of hard examples to the decision boundary is due to the entanglement of unhelpful style factors with useful content factors holds true, it would be logical to base the style factor on the natural style of images. This discrepancy needs to be addressed.

* It's unclear how this method compares to other existing methods in terms of computational efficiency. If CS-Isolate requires significantly more computational resources, it may not be practical for use in real-world applications.

* The authors did not discuss the limitations of the proposed method.

* Several sentences in the paper require further clarification, which will be discussed in the Questions section.

**Questions:**

* "If extracted examples have high quality, the performance of a classifier on them is enhanced.” This sentence is a bit confusing. Is the model trained on these examples?
* "However, During the training process, it is possible to identify and select some confident examples." It seems that "However" should be removed.
* "Some variation has also been proposed for example, some methods opt to reweight examples, thereby decreasing the contribution of mislabeled samples to the overall loss" This sentence could benefit from improved punctuation to enhance readability. Breaking it into smaller sentences or using punctuation to separate ideas could make the authors' point clearer and easier to understand.

**Limitations:**

The authors did not discuss the limitations of the proposed method.

---

> ### Author Rebuttal · Authors · 2023-08-10
>
> >**Q1. Support that hard examples are close to the decision boundary is due to the entanglement of style and content factors.**
>
> **A1**.
>  - **Visualization of Examples.** In Fig. 1 of Rebuttal PDF, We demonstrate examples that failed to be identified as confident by DivideMix but were successfully identified by our method. These examples often involve objects that are mixed with others, small, or located in corners.
>
> - **Grad-CAM Analysis of DivideMix.** We employ Grad-CAM to analyze the representation learned by DivideMix, and the results (also shown in Fig. 1 Rebuttal PDF) reveal that the corresponding heatmap highlights style parts of the images. This implies that DivideMix's learned representation contains style information that doesn't contribute usefully to the classification, causing the examples to lie closer to the decision boundary and then can not be selected as confident examples.
>
> - **Comparison with Our Proposed Method.**  We employ Grad-CAM to analyze content factors learned by CS-Isolate-DM (shown in Fig. 1 Rebuttal PDF). It shows that the content factors concentrate on the objects, effectively isolating content from unhelpful style factors that concentrate on other parts of the images. As a result, these examples are not hard examples anymore and are selected as confident examples by our method.
>
> >**Q2. How to design data augmentations that can manipulate all style factors in images.**
>
> **A2**.  As mentioned in Sec. 3 of our paper, in general, manipulating all style factors through data augmentations is considered unrealistic. Specifically, existing theoretical works [1] show that the disentanglement can be achieved if each style variable has a data augmentation to control its change. Although it claims that data augmentations can fully disentangle style, we've found that this assumption may be overly optimistic. Even with a rich set of augmentations, some style factors are typically uncontrolled. Therefore, we need additional auxiliary variables to further disentangle these uncontrolled style factors from content factors.
>
> Specifically, to further encourage disentanglement, we propose to leverage the labels of confident examples. Intuitively, when content labels are available, we can utilize them to control content factors, encouraging confident examples with the same label to share the same content factor. This process can further disentangle some uncontrolled style factors from learned content factors.
>
> To provide more insights, we have also formally analyzed this approach in Appendix A. If data augmentation is unable to control all style factors, it can only result in partial isolation, allowing for the separation of controlled style factors from content. To extend this separation to uncontrolled style factors, we can employ the labels of confident examples. For complete isolation of uncontrolled style factors and content factors, the existence of confident examples for all possible uncontrolled style factors is required.
>
> >**Q3. CS-Isolate focuses on learning style factors introduced through data augmentations, rather than modeling the natural style of an image.**
>
> **A3**. For most label-noise datasets, we do not have access to different style domain labels, making it hard to completely control the natural style of an image. Therefore, we propose using data augmentation as a surrogate to approximate style domains, achieving only partial isolation. To further encourage the disentanglement of content and style, we leverage labels from confident examples (See details in A2).
>
> It's also important to highlight that if datasets contain domain labels (whether labeled by humans or predicted by large models), they can be directly employed by our method to further enhance performance without any changes.
>
> >**Q4. Whether CS-Isolate requires significantly more computational resources.**
>
> **A4**. The additional computational resources are not a lot. For example, training 300 epochs on CIFAR-10 with batch size 64,  the training time of CS-Isolate-DM is approximately 21415s, and the training time of DivideMix is approximately 17667s; the GPU memory usage of DivideMix is 3457MiB, and the GPU memory usage of CS-Isolate-DM is 5007MiB; the number of parameters of DivideMix is 11171274, and the number of parameters of CS-Isolate-DM is 13250829.
>
> >**Q5. limitations of the proposed method.**
>
> **A5**. We will include limitations in the final version as follows.
>
> Manipulating all style factors through data augmentations is generally considered unrealistic, leading to a situation where content and styles are not completely disentangled but only partially disentangled. Achieving complete separation between uncontrolled style factors and content factors can be particularly challenging when learning with noisy labels. As a result, the confident examples selected in our method can encourage disentanglement but hard to guarantee the complete disentanglement of uncontrolled style factors from content factors.  Additionally, our approach requires additional computational resources to train extra encoder models.
>
> >**Q6. Is the model trained on extracted confident examples?**
>
> **A6**. All existing methods treat extracted confident examples as clean examples. Their noisy labels are used to train classifiers. Therefore, if the extracted confident examples have higher quality, the classifier trained on these examples can have higher classification accuracy.
> As a concrete example, Me-momentum [2] uses early stopping to select confident examples and train classifiers on the selected confident examples.
>
> ### References
>
> [1] Self-Supervised Learning with Data Augmentations Provably Isolates Content from Style, NeurIPS21.
>
> [2] Me-momentum: Extracting hard confident examples from noisily labeled data, ICCV21.

---

### Official Review · Reviewer_NWTZ · 2023-07-09

**Soundness:** 3 good
**Presentation:** 3 good
**Contribution:** 3 good
**Rating:** 6
**Confidence:** 4

**Summary:**

The problem of training deep neural networks with label noise is considered. Unlike traditional methods proposing to extract confident examples far from the classification boundary, this work proposes an approach named CS-Isolate to extract hard examples close to the classification boundary by disentangling content factor $Z_c$ and style factor $Z_s$. Different data augmentation techniques were employed to change the style factor $Z_c$ of images while the content factor remains unchanged $Z_s$. Two inference models were proposed to model the conditional distribution $Z_c |X$ and $Z_s |X$, respectively. Evaluation on synthetic noisy version of FashionMNIST, CIFAR-10, CIFAR-100 demonstrated the effectiveness of the proposed approach.

**Strengths:**

1.	To the best of my knowledge, this is the first work introducing fashion factor for extracting confident examples under label noise.
2.	The effectiveness of CS-Isolate was validated on the synthetic instance-dependent label noise.


**Weaknesses:**

1.	All evaluations were made on toy-level dataset CIFAR-10/100 and FashionMNIST and synthetic label noise. The effectiveness of CS-Isolate on large-scale dataset and real-world label noise (e.g., Clothing1M) is worth investigating.

**Questions:**

1.	In subsection section 2.2 (Sample selection methods for learning with noisy labels), it was worth discussing the idea to collect hard examples by active learning (e.g. [6]).

**Limitations:**

I didn't find a section dedicated to the discussion of limitations. One limitation in my mind is the lack of evaluation on real-world and large-scale label noise.

---

> ### Author Rebuttal · Authors · 2023-08-10
>
> We sincerely thank you for your time and effort to review our manuscript and please find the response to your comments below.
>
> >**Q1. The effectiveness of CS-Isolate on large-scale datasets and real-world label noise (e.g., Clothing1M) is worth investigating.**
>
> **A1**. The results on the Clothing1M dataset are in Tab. 4 of our Appendix B.3. We will move the results from Clothing1M to the main paper to emphasize the results on the real word noisy dataset. Additionally, we have also added the additional experiment of Cs-Isolate-Me and Cs-Isolate-Co. We follow the settings directly from their original paper. The experiment results are shown in Tab. 4 of Rebuttal PDF.
>
> >**Q2. In subsection 2.2 (Sample selection methods for learning with noisy labels), it was worth discussing the idea of collecting hard examples through active learning.**
>
> **A2**. Thanks for your insightful question. Firstly we would like to briefly introduce their method. Then we will analysis the connection between their methods and our methods.
>
> Intuitively, Jiacheng et al. [1] proposed using an estimated noise class posterior to select confident examples. Their method select the example with the high confidence of noise class posterior as confident examples. The reason they provide is that high confidence in the noisy class posterior likely corresponds to a small noise rate. Therefore, if we further assumes that easy-to-distinguish examples generally have a low noise rate, then Jiacheng et al.'s method can effectively select easy confident examples without using active learning.
>
> However, to select hard confident examples, they have to engage human annotators (active learning). Specifically, they randomly select some unconfident examples and manually label them. By contrast, we propose a method that can automatically learn some hard confident examples by leveraging disentangled representations. Our approach can be seen as an approximation of human labeling, and it does not require the intervention of human annotators.
>
> We would like to carefully sort this out in the final version.
>
> >**Q3. Discuss the limitations of the proposed method.**
>
> **A3.** Thank you for your advice. We will include limitations in the final version as follows.
>
> Manipulating all style factors through data augmentations is generally considered unrealistic, leading to a situation where content and styles are not completely disentangled but only partially disentangled. Achieving complete separation between uncontrolled style factors and content factors can be particularly challenging when learning with noisy labels. As a result, the confident examples selected in our method can encourage disentanglement but hard to guarantee the complete disentanglement of uncontrolled style factors from content factors. Additionally, our approach requires additional computational resources to train extra encoder models.
>
> ### Reference
>
> [1] Learning with bounded instance and label-dependent label noise, ICML20.

---

### Official Review · Reviewer_fCED · 2023-07-27

**Soundness:** 3 good
**Presentation:** 3 good
**Contribution:** 3 good
**Rating:** 6
**Confidence:** 4

**Summary:**

The paper proposed a novel method to extract the hard confidence examples under the noisy labels. By splitting the images into content factors and style factors, the hard confidence examples can be more easily extracted by solely considering the content factors. By training with the hard confidence examples, the models can achieve better generalization ability.

**Strengths:**

1. The paper presents a deep insight of the hard confidence example and why hard confidence examples are hard to extract.

2. A novel method is proposed in the paper to extract the hard confidence examples by splitting the examples into content factors and style factors. The two independent factors are produced by generative model with variational inference method. The experiments show that the proposed method achieves better performance than previous methods in the extraction of the hard confidence examples.

3. The proposed method shows better classification results than baselines.

**Weaknesses:**

1. Ablation study on hyper-parameters is missing.

2. The experiment datasets shown in the paper are relatively small dataset. Authors should provide experimental results on larger datasets such as Clothing1M. Since both baseline (DivideMix and Me-Momentum) mentioned in the paper report the results on this dataset. It is better to compare the results of the proposed methods against the baselines on this dataset.

3. Some errors in the writing. For instance. on line 276, the equation of conditional probability of style factors is incorrect. In eq.7, the minus sign should be plus sign.

**Questions:**

1. Can authors provide the visualization of the hard confidence examples as well as the content/style factors? It is good for readers to understand the effectiveness of the proposed method.

2. The paper just gives a rough description of the data augmentation techniques. Can author specify how many types of data augmentation used in the paper? (The value of $M$)? What is the difference between them? How the number of data augmentation techniques (value $M$) affects the disentanglement of content factors and style factors?

3. In eq. 3 authors mentioned about the refinement of content ID. How the authors deal with the conflicting of labels and content ID? e.g., sample whose label is not $p$ but assigned with content ID $p$.

**Limitations:**

Not applicable.

---

> ### Author Rebuttal · Authors · 2023-08-10
>
> >**Q1. Visualize the hard confident examples and the content/style factors.**
>
> **A1**. We visualize some examples that failed to be identified as confident examples by DivideMix, but are successfully identified by our method in Fig. 1 of Rebuttal PDF.
> We also visualize some examples that are successfully identified by both DivideMix and our method in Fig. 2 of Rebuttal PDF. It shows that objects in Fig. 1 are hard examples compared with objects in Fig. 2. Specifically, the objects in Fig. 1 are mixed with other objects, small, or located in the corners. The visualization implies that hard examples exist and our method can successfully identify these hard confident examples.
>
> In Fig. 3 of Rebuttal PDF, we also visualize Grad-CAM for content factors and style factors extracted by our proposed method, respectively. Grad-CAM of content factors mainly concentrates on the objects, in contrast, Grad-CAM of style factors mainly concentrates on other pixels on the images.
>
> Additionally, In Fig. 1 and 2 of Rebuttal PDF, we visualize Grad-CAM of the representation learned by DivideMix, which can be viewed as the mixture of content factors and style factors. It shows that for easy confident examples, DivideMix can identify the confident example successfully. The corresponding heatmap highlights semantic parts of the images. However, DivideMix fails to identify the hard confident examples and the corresponding heatmap highlights non-semantic parts of the images.
>
> The visualizations demonstrate that our method can disentangle content and style factors successfully and can help the existing method extract hard examples.
>
> >**Q2. Ablation study on hyper-parameters.**
>
> **A2**. We have conducted ablation studies on the dimension of $Z_c$ and $Z_s$, as well as the hyper-parameter $\lambda_{ELBO}$ on CIFAR-10N, and the noise type is ”Worst”. The experiment settings are as same as the description in Sec. 4.1 of our paper.
>
> **The ablation study on $\lambda_{ELBO}$**. We first fix the dimension of $Z_c$ and $Z_s$ on 32. Then change the value of $\lambda_{ELBO}$ to 0, 0.0001, 0.0005, 0.001, 0.002, 0.005 and 0.01, respectively. The experiment results are shown in Tab. 1 of Rebuttal PDF. The test accuracy increases gradually until $\lambda_{ELBO}$=0.001, then starts to decrease. The experiment results show that the $\lambda_{ELBO}$ should not be too large or too small. A preferred value of is near 0.001.
>
> **The ablation study on the dimension of $Z_c$ and $Z_s$**. with $\lambda_{ELBO}$ fixed on 0.001. The value of the dimension of $Z_c$ and $Z_s$ is 4, 8, 16, 32, 64, 128 and 256 respectively. The experiment results are shown in Tab. 2 of Rebuttal PDF. The test accuracy increases gradually until the dimension is 64. After the dimension is larger than 64, the test accuracy decreases slightly.
>
> >**Q3. Experimental results on Clothing1M.**
>
> **A3**. The results on the Clothing1M dataset are in Tab. 4 of our Appendix B.2. We will move the results from Clothing1M to the main paper to emphasize the results on the real word large noisy dataset. Additionally, we have also added the additional experiment of Cs-Isolate-Me and Cs-Isolate-Co. We follow the settings from their original paper. The results are shown in Tab. 4 of Rebuttal PDF.
>
> >**Q4. How many types of data augmentation were used?**
>
> **A4**. In our paper, we utilize 6 types of data augmentation techniques that follow [1].
>
> These techniques include shift scale rotation, random crop and horizontal flip, random brightness contrast, color jitter, and random to gray.  We generate different styles via the combination of different types of data augmentation. Each type of data augmentation has a probability to decide whether this augmentation will be applied. The probability of the random crop is 1; The probability of shift scale rotation, horizontal fip, random brightness contrast and random to gray is 0.5; The probability of color jitter is 0.8. Each combination acts as a style and will be assigned a unique style ID.
> We set the number of different data augmentation $M$ to be 200 in experiments.
>
> >**Q5. How does the number of data augmentation techniques affect the disentanglement?**
>
> **A5**. Existing theoretical works [1] show that disentanglement can be achieved if each style variable has a data augmentation to control its change.
> Although it posits that data augmentations can fully disentangle style, we've found that this assumption may be overly optimistic. Even with a rich set of augmentations, some style factors are typically uncontrolled. For example, in the CIFAR-10 dataset, horses often appear with people—an uncontrolled style factor that cannot be removed through data augmentation (Fig. 1 of Rebuttal PDF).
>
> Therefore, we emphasize using the labels of confident examples as content auxiliary variables to encourage the isolation of content factors from uncontrolled style factors. Intuitively, by ensuring that confident examples with the same label share the same content factor, we can disentangle uncontrolled style factors (see formal analysis in Appendix A).
>
> >**Q6. In the refinement process, how to deal with the conflicting labels and content ID?**
>
> **A6**. The content ID is initially randomly assigned. In the training process, we can obtain some high-quality confident examples with a precision exceeding 95% (as shown in Tab. 1 of our main paper). For these confident examples, we align their content IDs with the noisy labels, thereby refining the content IDs to accurately represent and be consistent with their clean labels. For unconfident examples, we let them have different content IDs and do not encourage their content factors to be the same.
>
> ### References
>
> [1] Self-Supervised Learning with Data Augmentations Provably Isolates Content from Style, NeurIPS21.

---

> > ### Comment · Reviewer_fCED · 2023-08-18
> >
> > Thanks for your clarification. My questions have been solved.

---

> > > ### Author Response · Authors · 2023-08-22
> > >
> > > Dear Reviewer fCED,
> > >
> > > Thanks for your appreciation to our work and your valuable comments.
> > >
> > > Best regards,
> > >
> > > Authors

---

### Author Rebuttal · Authors · 2023-08-10

Dear Reviewers,

**We sincerely thank you all for reviewing our paper and providing insightful feedback to help us improve our work.**

We are glad that the reviewers collectively acknowledge the paper's deep **insight into the extraction of hard confident examples** (Reviewer fCED, Reviewer NWTZ, Reviewer MtkC), addressing an **important and overlooked problem** in the field (Reviewer MtkC). The **newness, innovation, and novelty** of our methods have been appreciated, particularly those proposed for separating style factors and handling noisy labels (Reviewer fCED, Reviewer NWTZ, Reviewer MtkC). Our paper's method has been praised for its **effectiveness** in achieving better performance and new state-of-the-art results (Reviewer fCED, Reviewer NWTZ, Reviewer koUE, Reviewer 25vx). The writing quality was also commended, with specific compliments for the organization and **inclusion of visual aids** (Reviewer koUE).

We have tried our best to address all concerns and have uploaded Rebuttal PDF containing some exciting experimental results. The main contents of Rebuttal PDF are as follows:
- Visualization results of hard/easy confident examples, as well as the content/style factors;
- Ablation study on hyper-parameters;
- Experiment results on symmetric and pairflip noise;
- Experiment on Clothing1M.

**Once again, we thank all the reviewers for their time and effort in reviewing our paper. Please let us know if there are any further concerns.**

Sincerely,

Authors

---

### Author Response · Authors · 2023-08-12
**Discussion**

Dear Reviewers,

We are grateful for your time and the insightful feedback you have shared. In response to your comments, we have made every effort to address the concerns you raised and improve our work accordingly. Should you have any additional inquiries or need further clarification, please feel let us know.

Kind Regards,

Authors

---

> ### Author Response · Authors · 2023-08-13
> **Discussion**
>
> Dear Reviewers,
>
> Thanks for your efforts in reviewing this paper. We have tried our best to address the concerns and improve our work. Are there unclear explanations here? We can further clarify them.
>
> Best wishes,
>
> Authors

---

> ### Comment · Area_Chair_uitT · 2023-08-13
> **Questions for Authors**
>
> Dear Authors,
>
> One of the reviewers would like you to clarify some details on label noise levels. Thanks.
>
> Kind Regards,
>
> AC

---

> > ### Author Response · Authors · 2023-08-14
> > **Response**
> >
> > Dear SAC, AC, and Reviewers,
> >
> > Thanks for your comment. The detailed noisy level of dataset CIFAR-10N [1] is shown in the following table.
> >
> > | Worst  | Aggregate | Random 1 | Random 2 | Random 3 |
> > | ------ | --------- | -------- | -------- | -------- |
> > | 40.21% | 9.03%     | 17.23%   | 18.12%   | 17.64%   |
> >
> > To make it easier for readers to know the noise level of CIFAR-10N, we will add this table to the appendix.
> >
> > Best Regards,
> >
> > Authors
> >
> >
> >
> > ### Reference
> >
> > [1] Learning with Noisy Labels Revisited: A Study Using Real-World Human Annotations, ICLR22.

---

> > > ### Comment · Area_Chair_uitT · 2023-08-14
> > > **Response to Authors**
> > >
> > > Dear Authors,
> > >
> > > Thanks for the information. I will let you know if the reviewer has further questions.
> > >
> > > Kind Regards,
> > >
> > > AC

---

### Comment · Area_Chair_uitT · 2023-08-18
**Wait for Reviewers' Feedback**

Dear Reviewer NWTZ, koUE, and 25vx,

The authors and I are eager to know whether the author responses successfully address your concern. The Author-Reviewer discussion ends on **August 21**. You are strongly encouraged to directly reply to the authors.

Thank you for your hard work.

PS: (1) This is a public thread. (2) I am expecting to know your thoughts as well. If you want to individually reply to me, please use the thread of internal discussion.

Kind Regards,

AC

---

### Decision · Program_Chairs · 2023-09-21

**Decision:**

Accept (poster)

**Comment:**

I have read all the materials of this paper including the manuscript, appendix, comments, and response. Based on collected information from all reviewers and my personal judgment, I can make the recommendation on this paper, acceptance.

**Research Problem**

This paper considers a very specific research question on hard confident samples. Specifically, the existing literature regards the confident samples are far from the decision boundary, while the authors believe there exist some confident samples that are close to the decision boundary but hard to distinguish. In this paper, the authors aimed to extract these samples that are called them “hard confident samples.”

PS: 1. The authors use “examples” along with the manuscript, while I would like to use “samples.” 2. For some well-defined research question, I usually use one or two sentences for summarization. But for this paper, which targets a very specific research question, I have to restate some background.


**Motivation**

In Line 171-198, the authors clearly illustrated the research challenges. So, I omitted it here.

**Philosophy**

In Line 199-221, the authors demonstrated the philosophy to tackle the above challenges via data augmentation and self-learning. So, I omitted it again.

**Technique**

In general, the techniques in Section 3 make sense to me. A pseudo code is suggested in the Appendix.


**Experiments**

1. I agree with one reviewer that the current experimental results are reported in tables. It would be better to have some figures as well. For example, Figure 1 should be moved into this part.
2. Significant tests are suggested.
3. The authors need to report how many hard confident samples their algorithm detects and verify whether they are hard samples.

**Related Work**

I noticed a reference [1], which is not related to this paper in terms of research question but considers the content and style for a different task. I am not sure whether the authors are inspired by this paper. But it is better to discuss it in the manuscript.

[1] Harnessing Out-Of-Distribution Examples via Augmenting Content and Style, ICLR’23.

**Presentation**

Although the current version is readable and understood, the presentation can be further improved.

1. CS-Isolate in Line 16 is used without an introduction.
2. “During” in Line 87 -> “during.”
3. Figure 1 is the experimental result, rather than some evidence for illustrating the motivation. In a CS-style paper, this part should move to the experimental part.
4. Figure 2 is an illustrative example of motivation. However, more descriptions are needed; for example, what does the big red circle mean? What does blue and yellow mean?
5. “Challenging” in Line 171 should be “Challenges.”
6. “7”. In Line 184 should be “7.”
7. The layout of Figure 4 should not occupy Line 171. It is better not to occupy Line 158-159, either.
8. The roadmap before Section 3.1 is good.
9. It is better to provide informal definitions of hard/confident/noisy samples.
10. The intuition in Line 199-221 can be more concise.
11. $U_s$ and $U_c$ in Line 227 are used without an introduction.
12. There are many notations. If some notations are only used for 1-2 times, there is no necessary to give them a notation. Instead using their descriptions should be better.